# FRAME-VOYAGER: LEARNING TO QUERY FRAMES FOR VIDEO LARGE LANGUAGE MODELS

**Sicheng Yu**[1,*,†], **Chengkai Jin**[1,*], **Huanyu Wang**[1], **Zhenghao Chen**[1],
**Sheng Jin**[1], **Zhongrong Zuo**[1], **Xiaolei Xu**[1], **Zhenbang Sun**[1],
**Bingni Zhang**[1], **Jiawei Wu**[1,✉,†], **Hao Zhang**[2,✉], **Qianru Sun**[3]
[1]ByteDance   [2]Nanyang Technological University   [3]Singapore Management University

## ABSTRACT

Video Large Language Models (Video-LLMs) have made remarkable progress in video understanding tasks. However, they are constrained by the maximum length of input tokens, making it impractical to input entire videos. Existing frame selection approaches, such as uniform frame sampling and text-frame retrieval, fail to account for the information density variations in the videos or the complex instructions in the tasks, leading to sub-optimal performance. In this paper, we propose FRAME-VOYAGER that learns to query informative frame combinations, based on the given textual queries in the task. To train FRAME-VOYAGER, we introduce a new data collection and labeling pipeline, by ranking frame combinations using a pre-trained Video-LLM. Given a video of $M$ frames, we traverse its $T$-frame combinations, feed them into a Video-LLM, and rank them based on Video-LLM's prediction losses. Using this ranking as supervision, we train FRAME-VOYAGER to query the frame combinations with lower losses. In experiments, we evaluate FRAME-VOYAGER on four Video Question Answering benchmarks by plugging it into two different Video-LLMs. The experimental results demonstrate that FRAME-VOYAGER achieves impressive results in all settings, highlighting its potential as a plug-and-play solution for Video-LLMs.

## 1 INTRODUCTION

Recent studies (Liu et al., 2023; 2024a; Li et al., 2024c; Lin et al., 2024b) explore integrating Large Language Models (LLMs, Stiennon et al. (2020); Gao et al. (2023); OpenAI (2023); Touvron et al. (2023); Jiang et al. (2023); Yang et al. (2024)) with visual foundation models (*e.g.,* Vision Transformer (ViT, Dosovitskiy et al. (2021)) and cross-modal projectors (Wang et al., 2018; Lin et al., 2024a; Liu et al., 2023). In this paper, we focus on Video-LLMs. Existing Video-LLMs (Zhang et al., 2023a; Cheng et al., 2024; Li et al., 2024b) usually treat the video as a sequence of image frames. The key challenge is that the entire video can not be fed into the model due to LLMs' token length limitation (Xue et al., 2024). Meanwhile, arbitrarily increasing the token length of the model (Miao et al., 2023; Wan et al., 2024; Xiong et al., 2024; Zhang et al., 2024a) may lead to the "lost-in-the-middle" issue (Liu et al., 2024c) and introduce significant computational complexity.

To mitigate this issue, some efforts propose to select only a subset of frames as input, *e.g.,* through uniform sampling (Wang et al., 2019; Lin et al., 2024b; Cheng et al., 2024) or text-frame matching (Liang et al., 2024; Wang et al., 2024a; Yu et al., 2024). The uniform sampling strategy evenly samples frames in videos, while text-frame matching typically retrieves a set of relevant frames by calculating semantic similarities, *e.g.,* using CLIP (Radford et al., 2021), between the query and each frame. However, the uniform sampling fails to account for the information density variations in the videos. For instance, in the video question answering task, answering different questions may rely on distinct video segments or frames (Fu et al., 2024a). Meanwhile, text-frame matching is inadequate for complex video understanding tasks that require multi-frame or temporal reasoning, such as tracking the progression of an action or understanding cause-and-effect relationships over

---

[†]Project Lead.
[*]The first two authors contributed equally <{sicheng.yu,chengkai.jin2022}@bytedance.com>.
[✉]Correspondence to Jiawei Wu<wujiawei.viclan@bytedance.com> and Hao Zhang<hao007@e.ntu.edu.sg>.

time. For instance, in the video summarization task, simply matching frames might overlook the subtle transitions that connect scenes, while in temporal reasoning tasks—such as answering *why does the woman need to drink water at the beginning of the video?*—it is crucial to concentrate on the beginning part of the video. These matching-based methods fail to account for these frame-to-frame interactions and the relative positional information essential for a comprehensive understanding of the video. To solve these problems, we introduce an innovative approach named FRAME-VOYAGER that learns to query the subset of frames in a combinational manner, rather than retrieving individual frames separately. This capability is essential for understanding dynamic scenes and the global context of events.

To train FRAME-VOYAGER, we encounter two main challenges: **1) High Learning Complexity**: Learning the optimal combination of frames poses a combinatorial optimization problem. For instance, selecting 8 frames from a 128-frame video yields around $1.4 \times 10^{12}$ possible frame combinations. **2) Lack of Labeled Data**: There are currently no available datasets to facilitate the learning of such combinatorial problems in videos. We must address the question of how to construct a training dataset with minimal human effort. **To address the first challenge**, we formulate the combinatorial problem as a ranking task. Specifically, we train the model to rank the given frame combinations (*i.e.,* subsets) based on supervised ranking scores, which proves to be more efficient than forcing the model to search the optimal frame combination in a huge search space (Cao et al., 2007). Assume that for each frame combination, we have an annotation of ranking based on its usefulness in eventually generating the correct answer (by addressing the second challenge). Given a batch of frame combinations, the model learns to assign a higher reward to those with higher rankings. In other words, the model's objective is to maximize the reward for higher-ranked frame combinations. **To tackle the second challenge**, we propose leveraging a pre-trained Video-LLM to generate a ranking score for each frame combination, based on the prediction loss when the combination is input together with the query into this Video-LLM. The intuition is that a more effective frame combination will result in a lower prediction loss, indicating a higher likelihood of generating correct answers. Specifically, assume the original video has $M$ frames and the Video-LLM accepts only $T$ frames for input. We evaluate all frame combinations (*i.e.,* the total number is $\mathcal{C}(M, T)$) and rank them based on the prediction losses provided by the Video-LLM. The frame combinations with lower losses rank higher. These sorted frame combinations are then used for training FRAME-VOYAGER. Finally, the trained FRAME-VOYAGER is used for choosing a frame combination to input into Video-LLMs for downstream tasks.

To evaluate FRAME-VOYAGER, we plug it into two versions of the state-of-the-art Video-LLM named VILA (VILA-8B and VILA-40B, Lin et al. (2024b)) and conduct experiments on four widely-used Video Question Answering benchmarks, Video-MME (Fu et al., 2024a), MLVU (Zhou et al., 2024), NextQA (Xiao et al., 2021) and ActivityNet-QA (Yu et al., 2019). Experiment results show that using the frame combination "chosen" by FRAME-VOYAGER achieves significant performance improvements, compared to the conventional uniform sampling and text-frame retrieval (*i.e.,* individual text-frame matching) methods, especially for the cases requiring reasoning and information synopsis in long videos. **Our contributions** are thus two-fold: 1) We unveil the importance of combinational frame selection for video understanding tasks and propose an efficient method FRAME-VOYAGER that learns to do this frame selection automatically. The FRAME-VOYAGER itself is a plug-and-play module that can be applied to different Video-LLM architectures; 2) We formulate and learn FRAME-VOYAGER in a task of ranking frame combinations and introduce an automatic labeling pipeline to generate training datasets.

## 2 RELATED WORK

Transformer-based LLMs have revolutionized the field of natural language processing, achieving remarkable advancements by scaling up model sizes and expanding pre-training datasets (Brown et al., 2020; OpenAI, 2023; Chowdhery et al., 2023; Touvron et al., 2023). Researchers further extend LLMs into a multi-modality manner by fusing multi-modality information into the inputs of LLMs (Zhang et al., 2024b; Liu et al., 2023). In this work, we focus on Video-LLMs, and we consider the most widely-used structure (Lin et al., 2024b), where frames are adapted as visual tokens and then temporally fed into LLMs alongside text tokens in an auto-regressive way.

However, in existing Video-LLM models, a single frame is typically represented by $64$-$256$ visual tokens (Liu et al., 2023; Lin et al., 2024b; Cheng et al., 2024). Due to the input limitations of LLMs, the number of frames that can be processed by Video-LLM models is often constrained. Some studies apply techniques for handling long LLM inputs to support more frames (Miao et al., 2023; Wan et al., 2024; Xiong et al., 2024; Xue et al., 2024; Song et al., 2024), but this approach significantly increases computational complexity and can lead to issues such as the "lost-in-the-middle" effect and hallucinations (Liu et al., 2024c). Other studies, while keeping the frame input limit unchanged, use alternative sampling strategies instead of the default uniform sampling to obtain higher-quality frames as inputs. Some initial attempts focus on identifying transition frames (Lu & Grauman, 2013; Rochan et al., 2018; Rochan & Wang, 2019) or using frame clustering (Liang et al., 2024; Han et al., 2024) to find central frames, but these methods often overlook the information from the query. Subsequent research treats this problem as an individual text-frame (segment or cluster) matching task (Wang et al., 2024a; Yu et al., 2024; Wang et al., 2024b; 2025; 2024c), attempting to select frames that are semantically closest to the query. However, this method is sub-optimal as it ignores frame-to-frame relationships, failing in complex video understanding tasks requiring multi-frame or temporal information. An alternative way is to migrate the grounded video question answering (GVQA) methods (Xiao et al., 2024; Liu et al., 2025) to general video question answering. However, they focus on identifying specific continuous temporal segments directly related to a question (Zhang et al., 2020; 2022; 2023b), which cannot meet the requirements of general video question answering.

Therefore, to the best of our knowledge, our method is the first to consider the combination of frames as a whole, aiming to find the optimal combination that can best answer the query under the constraint of frame length limitations.

## 3 FRAME-VOYAGER

In the research context of Video-LLMs, typical video-language tasks such as video understanding, summarization, and reasoning can be formulated as video question answering tasks. The input and output of Video-LLMs are thus in the format of (*video*, *query*) and *answer*, respectively. The *video* here is typically not the entire video but rather a subset of frames, *i.e.,* frame combination, due to the token length limitations. Our research question is thus how to get the "optimal" subset of frames in this *video* to answer the *text query* correctly. Our method is called FRAME-VOYAGER. To train it with manageable complexity, we downsample the entire video to a fixed number of $M$ frames using uniform sampling. Then, we use FRAME-VOYAGER to evaluate $T$-frame combinations sampled from these $M$ frames, where $M \gg T$. The training is supervised and the labeled data are generated by a pre-trained reference Video-LLM, as elaborated in Section 3.1.

Given that the optimal combination of frames must be identified in a huge search space, one may wonder: *How is* FRAME-VOYAGER*'s training supervised?* We answer this question by formulating the problem as a ranking task (for which it is easy to get labeled data, *i.e.,* the second challenge), rather than looking for "optimal" (as it is non-trivial to learn, *i.e.,* the first challenge). In the following subsections, we elaborate on this task by introducing the pipeline of ranking data collection (Section (3.1)) as well as the training and inference processes of FRAME-VOYAGER (Section (3.2)).

### 3.1 DATA COLLECTION

We propose a human-free data collection and annotation pipeline for frame combinations. The overall process is demonstrated in Figure 1.

Our pipeline is based on a simple intuition: *if one frame combination is better than another, it will produce a lower language modeling loss when used as input to any trained Video-LLM*. Here we adopt loss instead of correctness as a metric, since correct answers may be generated based on the Video-LLM's inherent language prior without referring to any input content (Xiao et al., 2024). In contrast, loss reflects the model's confidence (Guo et al., 2017; Kadavath et al., 2022; Yu et al., 2025) in producing the answer given the frame combination and question, where lower loss indicates higher confidence. For each (*video*, *query*) pair, we evaluate all possible combinations of $T$ frames selected from a total of $M$ video frames, resulting in $\mathcal{C}(M, T)$ combinations, where $\mathcal{C}(M, T)$ is the binomial coefficient representing the number of ways to choose $T$ items from $M$. Each combination, along with the *query*, is then input into a trained reference Video-LLM to calculate the combination loss,

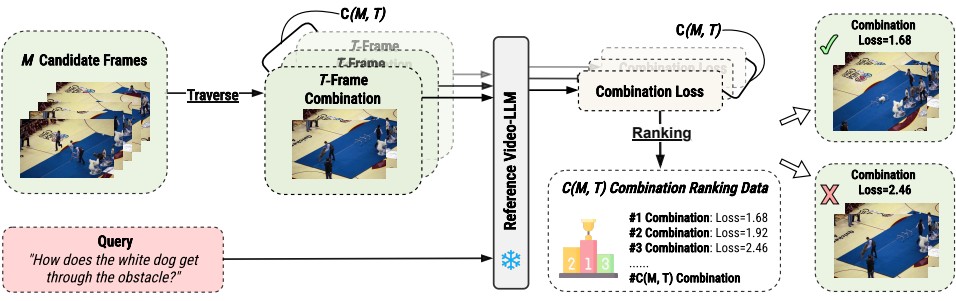

Figure 1: The data collection pipeline of FRAME-VOYAGER. Given a video of $M$ frames, we traverse its $T$-frame combinations, feed them into a Video-LLM, and rank them based on the reference Video-LLM's prediction losses. At last, we train FRAME-VOYAGER to query the frame combinations with lower losses. Please note that we omit filtering steps in this figure for clarity. $\mathcal{C}(M, T)$ is the binomial coefficient representing the number of ways to choose $T$ items from $M$.

*i.e.,* language modeling loss against the ground-truth *answer*. We collect the loss values for each combination and rank them from best to worst by sorting the losses in ascending order. It is worth noting that as $M$ increases, the potential number of combinations $\mathcal{C}(M, T)$ grows exponentially, making exhaustive traversal computationally infeasible. For example, when $M = 64$ and $T = 8$, the total number of combinations is approximately $\mathcal{C}(64, 8) \approx 4 \times 10^9$. Considering that the majority of the training data are with short videos, we use smaller combinations during training, such as $\mathcal{C}(16, 2)$ or $\mathcal{C}(32, 4)$. We observe from experiments that models trained with smaller combinations exhibit generalization capabilities when larger values of $M$ and $T$ are used during inference for longer video or complex reasoning. The specific choices of $M$ and $T$ employed in our experiments are detailed in Section 4.1.

To improve the ranking efficiency, we apply two filters for (*video*, *query*) pairs: we filter out 1) the pairs with an excessively high averaged loss, as these pairs may represent outlier cases or weak video-query correlations; and 2) the pairs with low variance in the losses across their combinations, as these pairs are not sensitive to the quality of combinations, *e.g.,* the correct answer may be generated solely based on the Video-LLM's inherent language prior without referring to any input content.

As a result, for each (*video*, *query*), we can obtain rankings for all $\mathcal{C}(M, T)$ frame combinations, *i.e.,* the combination ranking data in Figure 1. Each item in combination ranking data contains the indices of frames within the combination and its corresponding rank. For instance, given $M = 8$ and $T = 2$, the combination $Comb = \langle \{Frame^2, Frame^5\}, \#6 \rangle$ means that it contains the 2-th and 5-th frames from $M$ candidate frames, and it ranks at the #6 position given all traversed $\mathcal{C}(8, 2)$ combinations.

## 3.2 MODEL TRAINING AND INFERENCE

In this section, we elaborate on the training and inference details for FRAME-VOYAGER. Give a pre-trained Video-LLM, *e.g.,* VILA-8B (Lin et al., 2024b), we implement FRAME-VOYAGER as a lightweight plug-and-play module on it. For training this module, we use the $M$ candidate frames and query as input to the model, and our FRAME-VOYAGER is thus learned to capture both query-frame and frame-to-frame relationships. The overall process is demonstrated in Figure 2.

**Frame and Query Features.** Given uniformly sampled $M$ frames as candidate frames for each input (*video, query*) pair, we utilize the visual encoder followed by the Video-LLM projector to convert each frame into a sequence of visual tokens. Each visual token has the same size as the word token embedding of the LLM backbone. Thus, for $M$ frames, we have an initial feature map $\boldsymbol{X'} \in \mathbb{R}^{M \times N \times d}$, where $M$ is the number of frames, $N$ denotes the number of visual tokens per frame, and $d$ represents the feature dimension. We further perform token-wise average pooling before feeding them into LLMs for computational efficiency, *i.e.,* averaging $N$ visual tokens, on the initial frame feature map $\boldsymbol{X'} \in \mathbb{R}^{M \times N \times d}$ to obtain refined frame feature $\boldsymbol{X} \in \mathbb{R}^{M \times d}$. Concurrently, the query can be tokenized as $Q$ tokens, and filled with word embeddings of the backbone LLM. The operation will produce $\boldsymbol{Y} \in \mathbb{R}^{Q \times d}$ for textual information.

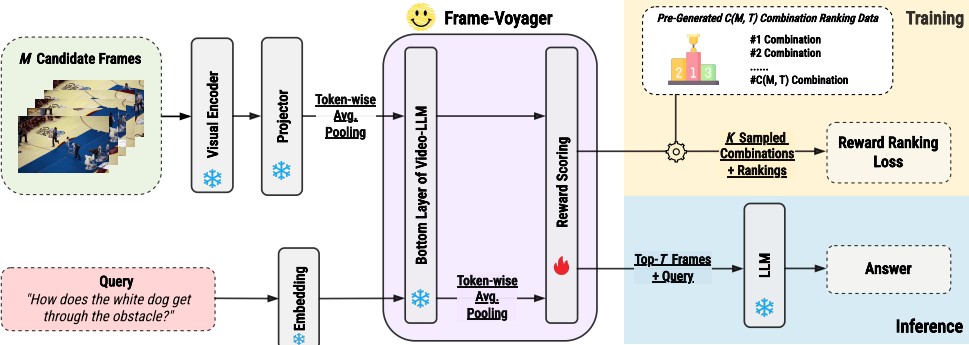

Figure 2: Training and inference processes of FRAME-VOYAGER. In the training, FRAME-VOYAGER is fed with all $M$ candidate frames and learns to rank $K$ sampled combinations from pre-generated combination ranking data in Section 3.1. Each combination contains $T$ frames. As for the inference, FRAME-VOYAGER selects top-$T$ frames with highest rewards to form the predicted frame combination. Note that there is no parameter to update during the inference.

**Cross-Modality Interaction Modeling.** To model both query-frame and frame-to-frame interactions, we leverage the bottom layers of LLMs. These transformer layers, which utilize the self-attention, are well pre-trained for vision-language tasks (Vaswani et al., 2017; Stan et al., 2024). Thus we concatenate the frame feature $\boldsymbol{X}$ and query feature $\boldsymbol{Y}$, and feed them together into LLMs' bottom layers. Importantly, all $M$ candidate frames are processed simultaneously, rather than individually feeding $T$ frames per combination, as FRAME-VOYAGER needs to model the entire set of $M$ candidate frames to capture the relationships within frames. The generated cross-attentive multimodal features are denoted as $\boldsymbol{X}_{\mathsf{BL}} \in \mathbb{R}^{M \times d}$ for frames and $\boldsymbol{Y}_{\mathsf{BL}} \in \mathbb{R}^{Q \times d}$ for the query. The BL refers for "Bottom Layer".

**Frame Combination Reward.** As mentioned, we formulate the combinatorial problem as a ranking task. Thus, for each frame combination, we need to compute its combination reward for further ranking-based training. First, we apply the token-wise average pooling on the cross-attentive multimodal features $\boldsymbol{Y}_{\mathsf{BL}}$ of query, and feed features into a feed-forward network (FFN). The generated final query feature is $\boldsymbol{Y}_{\mathsf{FFN}} \in \mathbb{R}^{h}$, where $h$ is the output dimension of the feed-forward network. The frame feature map $\boldsymbol{X}_{\mathsf{BL}}$ is also converted by another feed-forward network to get the final features $\boldsymbol{X}_{\mathsf{FFN}} \in \mathbb{R}^{M \times h}$. Then we simply measure the reward for a given frame combination as the averaged reward of the frames within the combination as:

$$r(\textit{Comb}) = \mathbb{E}_{i \in \textit{Comb}}[r(\textit{Frame}^i)]. \tag{1}$$

The reward for $i$-th frame (*i.e.,* $i$-th row in $\boldsymbol{X}_{\mathsf{FFN}}$) with respect to the query and other frames, is computed as:

$$r(\textit{Frame}^i) = \text{cosine}(\boldsymbol{Y}_{\mathsf{FFN}}, \boldsymbol{X}_{\mathsf{FFN}}^i). \tag{2}$$

**Training.** Inspired by the reward ranking loss function in (Ouyang et al., 2022), we train FRAME-VOYAGER via reward modeling to align its outputs with the optimal combination. To be specific, given the combination ranking data generated by the pipeline in Section 3.1, we uniformly sample $K$ ranked combinations, with each combination consisting of $T$ frames. Any two combinations sampled from the $K$ frame combinations are selected to form a training pair ($\mathcal{C}(K, 2)$ training pairs in total), with the frame combination having a lower loss in each pair designated as the chosen sample and the one with a higher loss as the rejected sample. Overall, given $K$ ranked combinations, the training objective, *i.e.,* reward ranking loss, is calculated as:

$$\mathcal{L} = -\frac{1}{\mathcal{C}(K, 2)} \mathbb{E}\left[ \log\left(\delta\big(r(\textit{Comb}_w) - r(\textit{Comb}_l)\big)\right) \right], \tag{3}$$

where $\textit{Comb}_w$ denotes the preferred frame combination, and $\textit{Comb}_l$ represents the rejected one.

Table 1: Comparing Video-LLMs with and without FRAME-VOYAGER as an additional module. Except for ours (**+FRAME-VOYAGER**) and the models with *, all results are copied from the related papers of benchmarks or models. The two VILA baselines utilize uniform sampling. For the Video-MME benchmark, we report results under two standard settings: without subtitles (no sub.) and with subtitles (sub.). ANQA refers to ActivityNetQA. Accuracy sign % is omitted for clarity.

| Model | LLM Size | Video-MME (no sub. / sub.) | | | | MLVU | ANQA | NextQA |
| | | Overall | Short | Medium | Long | | | |
| *Video Length* | | *17min* | *1.3min* | *9min* | *41min* | *12min* | *2min* | *0.8min* |
| Video-LLaVA | 7B | 39.9 / 41.6 | 45.3 / 46.1 | 38.0 / 40.7 | 36.2 / 38.1 | 47.3 | 45.3 | - |
| Qwen-VL-Chat | 7B | 41.1 / 41.9 | 46.9 / 47.3 | 38.7 / 40.4 | 37.8 / 37.9 | - | - | - |
| ST-LLM | 7B | 37.9 / 42.3 | 45.7 / 48.4 | 36.8 / 41.4 | 31.3 / 36.9 | - | 50.9 | - |
| VideoChat2 | 7B | 39.5 / 43.8 | 48.3 / 52.8 | 37.0 / 39.4 | 33.2 / 39.2 | 44.5 | 49.1 | - |
| Chat-UniVi-V1.5 | 7B | 40.6 / 45.9 | 45.7 / 51.2 | 40.3 / 44.6 | 35.8 / 41.8 | - | 46.1 | - |
| VideoLLaMA2 | 7B | 47.9 / - | 56.0 / - | 45.4 / - | 42.1 / - | - | 49.9 | - |
| LLaVA-NeXT-QW2 | 7B | 49.5 / - | 58.0 / - | 47.0 / - | 43.4 / - | - | - | - |
| LongVILA[128frm] | 8B | 49.2 / - | 60.2 / - | 48.2 / - | 38.8 / - | - | - | - |
| LongVILA[256frm] | 8B | 50.5 / - | 61.8 / - | 49.7 / - | 39.7 / - | - | - | - |
| VILA* | 8B | 47.5 / 50.0 | 57.8 / 61.6 | 44.3 / 46.2 | 40.3 / 42.1 | 46.3 | 53.7 | 55.6 |
| **+FRAME-VOYAGER** | 8B | 50.5 / 53.6 | 60.3 / 65.0 | 47.3 / 50.3 | 43.9 / 45.3 | 49.8 | 55.7 | 60.8 |
| LLaVA-One-Vision | 7B | 53.3 / - | 64.0 / - | 52.1 / - | 43.8 / - | 58.5 | 41.7 | 72.5 |
| **+FRAME-VOYAGER** | 7B | 57.5 / - | 67.3 / - | 56.3 / - | 48.9 / - | **65.6** | 48.4 | **73.9** |
| VideoLLaMA2 | 8×7B | 47.9 / 49.7 | - / - | - / - | - / - | - | 50.3 | - |
| VITA | 8×7B | 55.8 / 59.2 | 65.9 / 70.4 | 52.9 / 56.2 | 48.6 / 50.9 | - | - | - |
| LLaVA-NeXT-Video | 34B | 52.0 / 54.9 | 61.7 / 65.1 | 50.1 / 52.2 | 44.3 / 47.2 | - | **58.8** | - |
| VILA* | 34B | 58.3 / 61.6 | 67.9 / 70.7 | 56.4 / 59.8 | 50.4 / 52.1 | 57.8 | 56.8 | 62.9 |
| **+FRAME-VOYAGER** | 34B | **60.0 / 63.8** | **70.3 / 73.1** | **58.3 / 62.7** | **51.2 / 55.7** | 61.1 | 57.9 | 67.3 |

**Inference.** During inference, we plug the conventional Video-LLM models with our FRAME-VOYAGER module. Suppose that we uniformly sample $M$ candidate frames and expect $T$ frames as the visual input for Video-LLMs. The process remains consistent as the training procedure until the computation of the reward for frames. After that, we adopt the most efficient way to sample $T$ frames, *i.e.,* selecting the top $T$ frames with the highest rewards while maintaining their original temporal order in the video. The reason is that each reward of the frame here is optimized under combinational ranking supervision and incorporates the interaction information of the query and other frames.

## 4 EXPERIMENTS

### 4.1 EXPERIMENT SETTINGS

**Backbone Models.** We use three variants of the state-of-the-art Video-LLM, *i.e.,* VILA (Lin et al., 2024b): VILA-8B&40B (Wang et al., 2024b) and LLaVA-OneVision-7B (Li et al., 2024b). VILA-8B employs SigLIP (Zhai et al., 2023) as its visual encoder and Llama3-8B (Dubey et al., 2024) as its LLM backbone, while VILA-40B utilizes InternViT-6B (Chen et al., 2024) for visual encoding and Yi-34B (Young et al., 2024) as its LLM backbone.

**Training Data.** To ensure that FRAME-VOYAGER is trained on a diverse range of questions, we examine the video datasets used in VILA (Lin et al., 2024b) and LLaVA-OneVision (Li et al., 2024b). We select the training set of NextQA (Xiao et al., 2021) and VideoChatGPT (Maaz et al., 2024), on which we apply our proposed pipeline to create a training dataset for FRAME-VOYAGER. We empirically set the values of $M$ and $T$ based on the video length and question difficulty in these two datasets. Specifically, for NextQA, which features shorter videos and simpler questions, we select 16 candidate frames per video and explore all 120 possible 2-frame combinations. For VideoChatGPT, we select 32 candidate frames from each video and evaluate all $35,960$ possible 4-frame combinations. During the filtering process, we exclude the (*video, query*) pairs with an average loss larger than 7 and select pairs within the top 30% and 10% ranked by the variance of losses for the two datasets, respectively. After filtering, we obtain about $5,500$ and $7,000$ samples for these two datasets.

**Benchmarks.** We evaluate FRAME-VOYAGER on four widely-adopted video benchmarks: Video-MME (Fu et al., 2024a), MLVU (Zhou et al., 2024), NextQA (Xiao et al., 2021) and ActivityNet-QA (Yu et al., 2019). The former two evaluation datasets are tailored for long video assessments, while the latter two focus on short videos. We uniformly downsample the video to 128 (16) candidate frames and each time select 8 frames to compose a frame combination. The LMMs-Eval Library (Li et al., 2024a) is used for evaluation, and accuracy is reported across all benchmarks. Note that we report the accuracy scores of Video-MME under both without (no sub.) and with (sub.) subtitles settings.

Table 2: **RQ1.** Accuracies (%) for using different frame extraction methods on Video-MME (without subtitles). *Q*: whether query information is used. *Comb*: whether considering frame combination.

| | *Q* | *Comb* | Video-MME |
|---|---|---|---|
| VILA-8B (Uniform) | ✘ | ✘ | 47.5 |
| + RGB Histogram | ✘ | ✘ | 45.9 |
| + Edges Change Ratio | ✘ | ✘ | 47.3 |
| + Optical Flow | ✘ | ✘ | 46.7 |
| + Katna | ✘ | ✘ | 45.7 |
| + MDF | ✘ | ✘ | 47.8 |
| + TempGQA | ✔ | ✘ | 46.4 |
| + CLIP | ✔ | ✘ | 48.5 |
| + SigLIP | ✔ | ✘ | 48.3 |
| + InternViT-6B | ✔ | ✘ | 49.1 |
| + SeViLA | ✔ | ✘ | 49.3 |
| + VILA-Embedding | ✔ | ✘ | 48.8 |
| + FRAME-VOYAGER | ✔ | ✔ | 50.5 |

Table 3: **RQ2.** The ablation study (%) on different dataset collection methods. All results are evaluated on Video-MME (without subtitles). "Comb.": frame combination. In setting (4), The training is equal to predict the best frame combination with the lowest loss. In setting (5-6) and FRAME-VOYAGER (K=4), we adopt different numbers of combinations for ranking loss optimization in Equation 3.

| | Video-MME |
|---|---|
| (1) NextQA | 48.7 |
| (2) VideoChatGPT w/ Filtering | 49.1 |
| (3) VideoChatGPT w/o Filtering | 48.3 |
| (4) All Data + Top-1 Rank Comb. | 48.9 |
| (5) All Data + $K=2$ | 49.4 |
| (6) All Data + $K=8$ | 49.7 |
| FRAME-VOYAGER ($K=4$) | 50.5 |

**Implementation Details.** During the FRAME-VOYAGER dataset construction, VILA-8B is consistently adopted as the reference Video-LLM for generating loss due to resource limitations. During training, we set $K=4$ for sampling combination ranking data. VILA-8B is trained using Deep-Speed (Aminabadi et al., 2022) ZeRO2 with 8 H100 GPUs, while VILA-40B is trained using ZeRO3 setting with 32 H100 GPUs. The batch size (with accumulation) is set to 64 and the learning rate is $1e^{-3}$. The training of FRAME-VOYAGER is conducted over 40 epochs requiring approximately 8 hours for VILA-8B whereas over 20 epochs for VILA-40B, taking around 20 hours. All model inferences are performed on 8 H100 GPUs.

## 4.2 RESULTS AND ANALYSIS

**Comparison with state-of-the-art methods.** Table 1 presents the comparison of FRAME-VOYAGER with leading Video-LLMs, organized by the size of their backbone LLMs. For models with LLMs of 8B parameters or fewer, we evaluate Video-LLaVA (Lin et al., 2024a), Qwen-VL-Chat (Bai et al., 2023), ST-LLM (Liu et al., 2024d), VideoChat2 (Li et al., 2023), Chat-UniVi-V1.5 (Jin et al., 2024b), VideoLLaMA2 (Cheng et al., 2024), LLaVA-NeXT-QW2 (Liu et al., 2024b), LLaVa-OneVision (Li et al., 2024b) and LongVILA (Xue et al., 2024). For larger models, we compare FRAME-VOYAGER with VideoLLaMA2 (Cheng et al., 2024), VITA (Fu et al., 2024b) and LLaVA-NeXT-Video (Zhang et al., 2024c).

Among the models with LLMs of 8B parameters or fewer, FRAME-VOYAGER achieves the best overall performance. On the Video-MME benchmark (without subtitles), it outperforms the vanilla VILA-8B by 3.0%, with a notable 3.6% gain on long videos. Remarkably, FRAME-VOYAGER, using only 8 frames as input into VILA, surpasses the VILA variant LongVILA, which utilizes 128 and 256 frames and requires additional training and system design. LongVILA improves its performance by inputting more frames, while FRAME-VOYAGER improves through querying more informative frame combinations, without changing the frame length limits of VILA. FRAME-VOYAGER outperforms LongVILA even on extremely long videos (average 41 minutes). This suggests that simply increasing the input number of frames may not always lead to better performance, since incorporating more

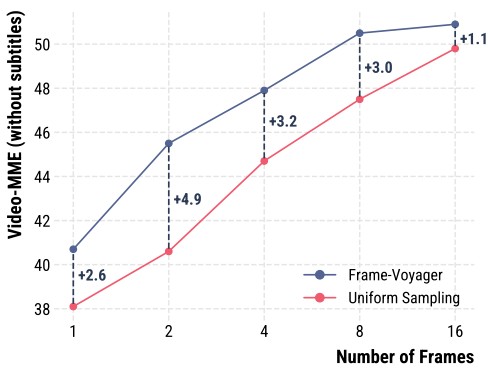 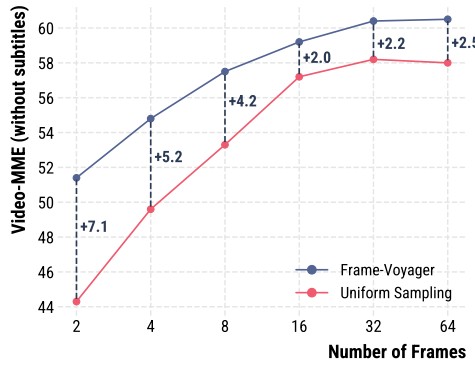

(a) Video-LLM Backbone: VILA-8B.    (b) Video-LLM Backbone: LLaVA-OneVision-7B.

Figure 3: Accuracies (%) of uniform sampling and FRAME-VOYAGER on Video-MME (without subtitles) regarding number of frames. Both models use the same number of candidate frames (128).

frames might introduce noises and irrelevant information. More frames also reduce computing efficiency. Besides, among the larger Video-LLMs (with 8×7B and 34B LLM backbones), FRAME-VOYAGER consistently brings notable improvements over the vanilla VILA and other state-of-the-art models. These results highlight the importance of selecting and utilizing the optimal information from video for efficient video understanding. Last but not least, VILA-8B and VILA-40B employ distinct vision encoders and LLM backbones, as outlined in Section 4.1. Thus, the consistent performance improvements indicate that FRAME-VOYAGER functions as a plug-and-play module compatible with different Video-LLM architectures. Results on additional benchmarks (MVBench (Li et al., 2024d), STAR (Wu et al., 2021), and EgoSchema (Mangalam et al., 2023)) can be found in Appendix C.

**Research Question (RQ) 1: How does FRAME-VOYAGER compare to other frame extraction methods?**

To evaluate the effectiveness of FRAME-VOYAGER on Video-LLMs, we conduct experiments with several baseline methods, all utilizing the same VILA-8B backbone and the same number of frames for a fair comparison. We test rule-based shot boundary detection (SBD) methods, including Histogram (Sheena & Narayanan, 2015), Edges Change Ratio (Nasreen & Dr Shobha, 2013), Motion (Wolf, 1996), and MDF (Han et al., 2024), which select frames based on significant transitions in texture, structure, motion and inherent similarity. Frames with the most substantial changes are chosen as extracted frames. We also include Katna[1], a cluster-based method that extracts histograms from all frames and uses K-means clustering to select most representative frames near cluster centers. In addition, six frame-text matching methods (Liang et al., 2024; Wang et al., 2024a), including VILA-Embedding, CLIP (Radford et al., 2021), SigLIP (Zhai et al., 2023), InternViT-6B (Chen et al., 2024), TempGQA (Xiao et al., 2024) and SeViLA (Yu et al., 2024), are employed to retrieve frames by calculating cosine similarity between query inputs and individual frames. Implementation details are presented in Appendix A.

Table 2 presents the comparison results on Video-MME (without subtitles). Rule-based methods perform worse than uniform frame sampling, likely due to inherent biases in these techniques. For instance, optical flow methods prioritize motion-heavy frames, while RGB histogram methods emphasize texture changes. These approaches overlook the input *query* and thus often fail to answer the *query*. Furthermore, although VILA Embedding and CLIP outperform uniform sampling, they still fall short compared to FRAME-VOYAGER. Their frame-by-frame extraction approach lacks a holistic understanding of the video, making them struggle with complex tasks requiring temporal reasoning and comprehensive video comprehension. Overall, the proposed FRAME-VOYAGER outperforms all the baselines, demonstrating superiority for frame subset selection and efficient video understanding.

**RQ2: What is the impact of each component on data collection?**

To evaluate the effectiveness of each component in our dataset construction phase, we conduct a series of ablation studies summarized in Table 3. Specifically, we investigate the impact of individual

---

[1] https://github.com/keplerlab/katna

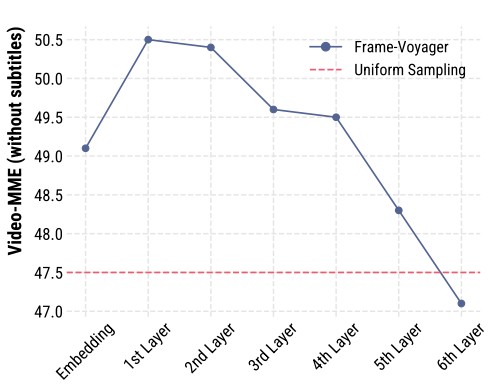 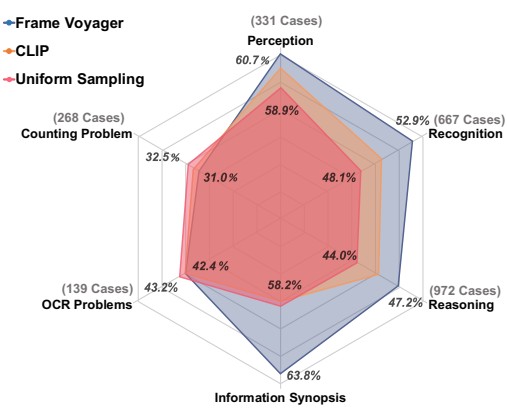

Figure 4: **RQ4.** Performance of FRAME-VOYAGER reusing different parts of VILA-8B on Video-MME (without subtitles).

Figure 5: **RQ5.** Performance of FRAME-VOYAGER, CLIP, and uniform frame sampling on six question types of Video-MME.

datasets (1-2), analyze the role of data filtering (2-3), and explore the effects of different data usage strategies during training (4-6). In (4), we directly optimize the FRAME-VOYAGER by the combination ranked highest. In (5-6), we modify $K$, number of sampled combinations, during training. The results of (1-2) demonstrate additive performance gains from each dataset and the significant distribution differences between the two datasets enhance the query diversity. The results of (3) underscore the critical role of data filtering in eliminating instances unsuitable for training FRAME-VOYAGER. Experiments (4-6) provide some insights on data usage during training, revealing that varying the number of combinations can adversely affect reward computation. Moreover, results from (4) highlight the necessity of teaching FRAME-VOYAGER to discern better from worse combination via reward modeling, rather than merely training it to identify the combination.

**RQ3: How does the number of frames impact the performance of FRAME-VOYAGER?**

In Figure 3, we demonstrate the performance of FRAME-VOYAGER on the Video-MME (without subtitles) by varying the number of chosen frames and comparing it to uniform sampling. Across different numbers of frames, FRAME-VOYAGER consistently outperforms uniform sampling. Notably, FRAME-VOYAGER achieves better results using only half the frames, *e.g.,* the 8-frame FRAME-VOYAGER surpasses the 16-frame uniform sampling. However, as the number of extracted frames increases, the performance gap between FRAME-VOYAGER and uniform sampling narrows. There are two primary factors that influence the performance. First, public benchmarks typically do not require a large number of frames to answer queries. Incorporating more frames may introduce unnecessary information and potentially limit performance gains. Second, performance is constrained by the model's capabilities. Given a selected frame combination, the inference model, *e.g.,* the frozen VILA, determines the performance's upper bound. For instance, Figure 5 demonstrates that VILA generally underperforms on OCR and counting tasks.

**RQ4: How does the design of FRAME-VOYAGER contribute to the performance?**

FRAME-VOYAGER reuses the embedding layer and the first transformer layer of the backbone LLM. To validate this design choice, we assess the impact of reusing different components of the LLM, as presented in Figure 4. Reusing only the embedding layer's bag-of-words features yields noticeable improvements. Incorporating one to two layers of the LLM enhances question understanding, leading to better performance. However, reusing additional layers results in a gradual performance decline. This decline occurs because the LLM remains frozen during FRAME-VOYAGER's training, while lower-layer features are general-purpose, higher layers—with increased attention and fusion operations—shift the features towards language modeling, leading to diminished performance (Jin et al., 2024a).

**RQ5: How does FRAME-VOYAGER perform on different types of questions?**

Leveraging the question types defined in the Video-MME benchmark, we conduct a comparative analysis among three methods, *i.e.,* our FRAME-VOYAGER, the uniform sampling approach, the

individual frame-query matching method based on CLIP, across six distinct categories of questions. The results are presented in Figure 5, where the maximum and minimum values are attached to each type in the figure. The results indicate that FRAME-VOYAGER achieves significant improvements in four of these categories, including an accuracy enhancement of $4.8\%$ on the recognition task compared to uniform sampling. However, slight performance fluctuations are observed in the counting and OCR tasks. For instance, FRAME-VOYAGER results in one additional error in the counting problem compared to uniform sampling. We attribute these minor inconsistencies to VILA's inherent limitations in effectively handling these specific types of tasks.

We can also observe that, although CLIP-based method shows improvement over uniform sampling in the perception, recognition, and reasoning question types, they still fall short compared to FRAME-VOYAGER. In the information synopsis type, CLIP-based method performs even worse than uniform sampling. The reason is that CLIP-based methods ignore the global video information, whereas FRAME-VOYAGER explicitly models both the query-frame and frame-to-frame interactions.

**RQ6: What does the combination extracted by FRAME-VOYAGER look like?**

In Figure 9 of Appendix I, we present one sampled case from Video-MME (Fu et al., 2024a). The (*video, query*) pair, with the query *"What is the small flying black dot at the start of the video"*, is evaluated across different methods. The frames obtained using uniform sampling provide limited information, with irrelevant background frames included. When using CLIP for frame-query matching, the retrieved frames show a small black dot in the first frame, followed by frames related to the terms "virus" and "protein". However, these frames do not clarify what the small black dot represents.

In contrast, our model, FRAME-VOYAGER, effectively queries frame combinations by analyzing relationships between frames and modeling temporal information. This enables FRAME-VOYAGER to capture the critical details at the start of the video. The selected frame combinations reveal the trajectory of the "small black dot in flight", ultimately identifying the "dot" as a "virus". Additional two case studies are provided in Figure 10 and Figure 11 of Appendix I.

**RQ7: How does FRAME-VOYAGER maintain efficiency and scalability when processing an increased number of selected frames?**

As mentioned in Section 4.1, when training, we sample 4(2) frames from 32(16) frames, and when inference, we directly choose 8 frames from 128 frames. This demonstrates that our FRAME-VOYAGER can be generalized from smaller numbers to larger numbers of selected frames. It is aligned with most of the existing Video-LLMs, i.e., trained primarily on short videos, yet they can also generalize to long videos during testing (Lin et al., 2024b; Li et al., 2024b). Additionally, we use two pruning methods to address efficiency concerns and reduce computational resource consumption of the data collection pipeline in Section 3.1. The results show that the data construction time can be reduced to only $4.4\%$ of the full version, while maintaining comparable performance across all benchmarks. The pruning details can be found in Appendix G.

## 5 CONCLUSIONS

In this work, we introduce FRAME-VOYAGER, a plug-and-play frame combination selection method that enhances Video-LLMs in video understanding tasks. We address the challenges of combination optimization by formulating it as a ranking task and implementing a ranking-based reward learning framework with a human-free data collection pipeline. Extensive experiments show that FRAME-VOYAGER not only significantly boosts the performance of baseline Video-LLMs like VILA, achieving state-of-the-art results, but also outperforms other frame selection methods. Comprehensive ablation studies further confirm its effectiveness. Overall, our work sets a strong baseline for Video-LLMs and guides future research on frame selection and optimization. As Video-LLMs evolve to tackle diverse tasks, our method offers an efficient solution to enhance broader video understanding.

## LIMITATIONS

For the data construction, due to resource constraints, 1) we only generate the ranking data using VILA-8B, precluding experiments with more powerful Video-LLMs that might yield superior results.

2) The combinations for data construction are limited in size as the training set primarily focuses on short videos. Despite the promising improvements achieved, we highlight that applying FRAME-VOYAGER to longer videos with a greater number of frame combinations could potentially lead to further performance enhancements in processing extended video content. For the model framework, 1) we integrate our approach into existing Video-LLMs as a plug-in to ensure parameter efficiency, without additional fine-tuning of the backbone model. However, directly reusing the backbone's parameters may not yield optimal results. Moreover, our simplified plug-in module could benefit from a more sophisticated design to further enhance overall performance. 2) Integrating our approach into the pre-training process of Video-LLMs remains unexplored, we hypothesize that learning frame combinations during pre-training could produce a more robust and effective model.

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

## A    IMPLEMENTATION DETAILS OF FRAME EXTRACTION BASELINES IN RQ1

In executing the SBD methods, we utilize the OpenCV[2] library. We generate the disparity between each frame and its adjacent frame, utilizing both the difference of the RGB Histogram and Canny, as well as the optical flows. This process aids us in picking out $T$ frames with the top-$T$ greatest disparity values.

For the cluster-based approach, Katna, we directly use Katna API. It initially segments the videos according to content specifics and extracts a list of keyframes from each individual segment, employing the Histogram with K-means technique. Afterward, all segmented keyframe lists are merged to pick out the final $T$ frames. For MDF (Han et al., 2024), we implement it utilizing the official code with default parameters[3]. To compute the similarities between frames, we utilize the features from VILA-8B's visual encoder.

For TempGQA (Xiao et al., 2024), we select a specific grounding segment based on the question following the official code[4]. Then we uniformly sample frames from the selected segment and feed them into VILA-8B with questions to generate answers.

For SeViLA (Yu et al., 2024), we utilize its frame selection module, *i.e.,* the localizer in SeViLA, and maintain the original hyperparameter settings[5].

For VILA-Embedding, we utilize the visual feature after the projector and the text feature from word embedding to measure the cosine similarity after the average pooling. The cosine similarity is then used for ranking the frames. For CLIP[6] (Radford et al., 2021), SigLIP[7] (Zhai et al., 2023), and InternViT-6B (Chen et al., 2024), we directly rank and select top-$T$ candidate frames according to their output logits.

## B    QUESTION TYPE ANALYSIS OF THE GENERATED DATASETS

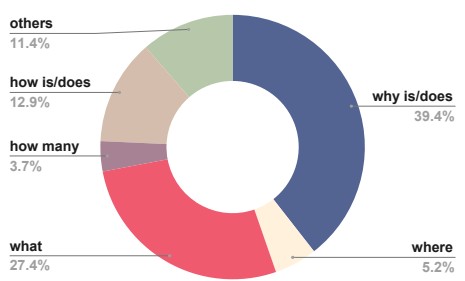

(a) Question type of origin dataset.

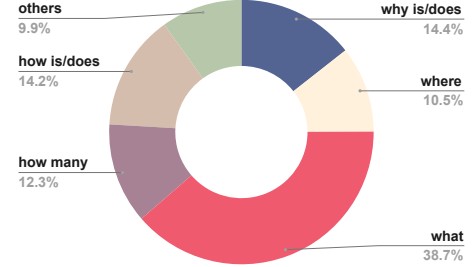

(b) Question type of our dataset.

Figure 6: The question type distribution on NextQA dataset. Best viewed in color.

We observe similar patterns in Figure 6 and Figure 7. Taking the NextQA dataset as an example, the proportion of "what" type questions in the original dataset is 27.4%, whereas in our generated dataset, this proportion increases to 38.7%. Conversely, the proportion of "why is/does" type questions significantly decreases from 39.4% in the original dataset to 11.4% in our generated dataset.

The reason is that answers to questions starting with "what" tend to be shorter compared to those starting with "why is/does". When we calculate the loss for each subset combination using Video-LLM, the auto-regressive loss for each token in the answer is computed based on all preceding tokens. In the case of a long answer, the LLM's prior knowledge may diminish the impact of the subset

---

[2]https://github.com/opencv/opencv
[3]https://github.com/declare-lab/Sealing
[4]https://github.com/doc-doc/NExT-GQA/tree/main/code/TempGQA
[5]https://github.com/Yui010206/SeViLA?tab=readme-ov-file
[6]https://huggingface.co/openai/clip-vit-large-patch14-336
[7]https://huggingface.co/google/siglip-so400m-patch14-384

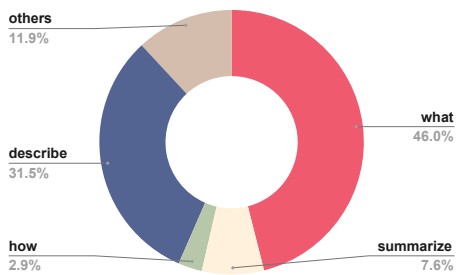

(a) Question type of origin dataset.

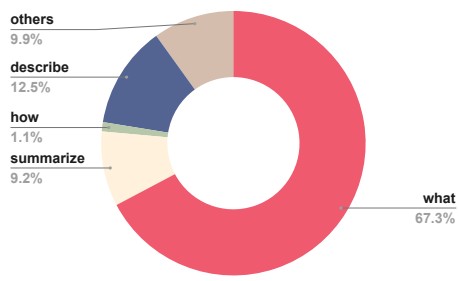

(b) Question type of our dataset.

Figure 7: The question type distribution on VideoChatGPT dataset. Best viewed in color.

combination (as the LLM can predict subsequent tokens based on the earlier ones in the answer). This leads to more "why" cases being removed during our filtering process.

## C   ANALYSIS OF COMPUTATION COST

In this section, we analyze the additional overhead introduced by FRAME-VOYAGER based on VILA-8B. First, in terms of parameter size, our method introduces two MLPs, and the additional parameters account for $0.2\%$ of the original model's parameter. Next, we analyze the time overhead. During training, the original model required $5.1k$ GPU hours(Lin et al., 2024b), while FRAME-VOYAGER requires $64$ GPU hours, representing an additional training time of $1.25\%$ of the original.

For inference, we randomly sample $100$ examples from Video-MME to measure the model's inference latency. Using the experimental setting from our main paper, the average inference latency for the uniform sampling baseline is $1.329$ seconds per example, while for FRAME-VOYAGER it is $1.696$ seconds, indicating that our method introduces an additional latency overhead of approximately $27.6\%$. Meanwhile, Frame-Voyager uses $23,069$ MiB and the baseline uses $22,425$ MiB, making a difference of $2.9\%$.

## D   RESULTS ON MORE BENCHMARKS

In this section, we present additional results on three additional benchmarks, *i.e.,* MVBench (Li et al., 2024d), STAR (Wu et al., 2021), and EgoSchema (Mangalam et al., 2023), in Table 4. The results show that our method consistently outperforms uniform sampling on the first two benchmarks, while the improvement of FRAME-VOYAGER on EgoSchema is marginal. After carefully examining the EgoSchema dataset, we identify the main reasons are ambiguous pronouns and camera wearer (cameraman) related questions. Upon 50 random sampled instances from EgoSchema, we observe that 44 instances contain the character "c" rather than the conventional pronoun to represent the human. For example, "what was the primary tool used by c in the video" and "what can be inferred about c's assessment of the plants during the video". We find that "c" represents the "camera wearer", who may not even appear in the video. Such special characteristics impede Frame-Voyager's ability to identify truly query-relevant frame combinations.

Table 4: Results on MVBench, STAR and EgoSchema. Accuracy sign $\%$ is omitted for clarity.

|  | MVBench | STAR | EgoSchema |
| --- | --- | --- | --- |
| VILA-8B | 40.1 | 48.3 | 53.3 |
| **+FRAME-VOYAGER** | 41.1 | 50.5 | 53.6 |
| VILA-40B | 56.2 | 62.1 | 63.2 |
| **+FRAME-VOYAGER** | 57.3 | 63.8 | 63.3 |

# E    A Study on the Number of Candidate Frames

Table 5: The ablation results on different number of candidate frames. For all experiments, we expand the candidate frames from 8 to 256, while freeze the number of chosen frames as 8. Results are reported on Video-MME dataset (without subtitles).

| #candidate frames | 8 | 16 | 32 | 64 | 128 | 256 |
|---|---|---|---|---|---|---|
| Video-MME (%) | 47.5 | 48.2 | 48.6 | 49.7 | 50.5 | 50.8 |

As the number of candidate frames increases, more video information is captured within the selection pool. While selecting 8 frames from a larger candidate set, our method consistently improves results on the Video-MME dataset. As shown in Table 5, selecting 8 frames from a candidate pool of 128 frames yields a 3% improvement compared to selecting from just 8 frames (*i.e.,* uniform sampling). However, as the candidate set size increases further (from 128 to 256), the additional information brought by the expanded set becomes limited.

# F    Dynamic Frame Selection

In this section, we further explore the capability of FRAME-VOYAGER on dynamic frame selection. Specifically, we modify the inference process by normalizing the rewards of candidate frames ranging from 0 to 1. Then we only select the frames that exceed a specified threshold. We conduct experiments on Video-MME (without subtitles) using VILA-8B and LLaVA-OneVision-7B. For VILA-8B, we set a normalized reward threshold of 0.7 and maintain our original constraints (maximum 8 frames, minimum 1 frame). For LLaVA-OneVision-7B, we expand the maximum frame limit to 32 and adjust the threshold to 0.5. The overall results is shown in Table 6.

Building on the experiment with VILA-8B, we further examine the required information density (defined by the number of dynamically selected frames) from three perspectives: video duration, video content domains, and query types. We present our experimental results below in Table 7, Table 8 and Table 9. The results reveal several patterns about how FRAME-VOYAGER selection adapts to different scenarios:

- **Video Duration**: Longer videos naturally require more frames (Short: 3.72, Medium: 4.07, Long: 4.61 frames on average).

- **Video Content Domains**: The correlation between frame requirements and video domains is less obvious. Sports videos require the most average frames (4.43) due to frequent action changes, while other domains maintain relatively consistent frame requirements (3.89-4.31).

- **Query Types**: Task complexity does influence frame selection. Complex tasks like counting (5.36 frames) and reasoning (4.58 frames) require more frames for comprehensive analysis. Note that counting tasks, despite having lower reasoning demands, usually requires answering questions like "How many different people appear in the video?", which strongly depend on multiple frames. Simpler tasks like perception (3.01 frames) and recognition (3.84 frames) need fewer frames.

The results confirm FRAME-VOYAGER's effectiveness in dynamically selecting frames by considering both the video's information density and the specific query needs.

# G    Optimization of Data Collection Pipeline

We implement two straightforward approaches to address efficiency concerns and reduce computational resource consumption in the data collection pipeline.

Table 6: The results with different sampling methods on Video-MME dataset (without subtitles). Averaged numbers of frames are reported and accuracy sign % is omitted for clarity.

| Video-LLM Backbone | Method | Avg. Frames | Acc. |
|---|---|---|---|
| VILA-8B | Uniform Sampling | 4 | 44.7 |
| | FRAME-VOYAGER + Fixed Num. | 4 | 47.9 |
| | Uniform Sampling | 8 | 47.5 |
| | FRAME-VOYAGER + Fixed Num. | 8 | 50.5 |
| | **FRAME-VOYAGER + Dynamic Num.** | 4.1 | 49.6 |
| LaVA-One-Vision-7B | Uniform Sampling | 8 | 53.3 |
| | FRAME-VOYAGER + Fixed Num. | 8 | 57.5 |
| | Uniform Sampling | 16 | 57.2 |
| | FRAME-VOYAGER + Fixed Num. | 16 | 59.2 |
| | Uniform Sampling | 32 | 58.2 |
| | FRAME-VOYAGER + Fixed Num. | 32 | 60.4 |
| | **FRAME-VOYAGER + Dynamic Num.** | 13.3 | 59.1 |

Table 7: Averaged numbers of frames for different lengths of videos. The analysis is conducted based on the results of VILA-8B in Table 6.

| Video Duration | Avg. Frames |
|---|---|
| Short | 3.72 |
| Medium | 4.07 |
| Long | 4.61 |

Table 8: Averaged numbers of frames for different video content domains. The analysis is conducted based on the results of VILA-8B in Table 6.

| Video Content Domain | Avg. Frames |
|---|---|
| Knowledge | 3.89 |
| Film & Television | 3.96 |
| Sports Competition | 4.43 |
| Artistic Performance | 3.95 |
| Life Record | 3.97 |
| Multilingual | 4.31 |

Table 9: Averaged numbers of frames for different types of questions. The analysis is conducted based on the results of VILA-8B in Table 6.

| Video Duration | Avg. Frames |
|---|---|
| Perception | 3.01 |
| Recognition | 3.84 |
| Reasoning | 4.58 |
| OCR | 3.93 |
| Counting | 5.36 |
| Information Synopsis | 4.35 |

**Method (1).** We conduct dynamic pruning by filtering combinations based on frame-to-frame similarity and temporal proximities among frames. This strategy helps us discard the frame combinations containing multiple similar and redundant frames. It reduces the average number of combinations per sample to 14% on the VideoChatGPT dataset with processing time reduced to about 5 hours using 32 A100 GPUs and to 27% on the NextQA dataset with about 15 minutes using 8 A100 GPUs.

**Method (2).** We further utilize CLIP Radford et al. (2021) to compute the similarity between all frames and the question, and rank them accordingly. We mark lower-ranked frames as irrelevant. We believe that only a small portion of video frames directly relate to the question, while the majority of frames are irrelevant. As a result, most frame combinations only containing irrelevant frames can be discarded during the training process. Based on this pruning strategy, we filter out most combinations consisting of only irrelevant frames while retaining a few as low-ranking training samples. This approach can further reduce data collection time by 60-70%.

The comparison among the vanilla data collection method, +Method (1), and +Method (1) & (2) is shown below in Table 10. The backbone model is kept as VILA-8B. The results demonstrate that the data collection time of FRAME-VOYAGER could be largely reduced to just 4.4% of the original version, while maintaining comparable performance across all benchmarks.

Table 10: The comparison between the vanilla data collection method with the optimized methods. GPU Hours consists of two parts: the former is the time used on VideoChatGPT dataset and the later that of NextQA dataset. The time costs show the GPU Hours saved by the optimization methods. Accuracy sign % is omitted for clarity.

|  | GPU Hours | Time Costs | Video-MME | MLVU | NextQA | ANQA |
|---|---|---|---|---|---|---|
| Vanilla Method | 1280+8 | 100% | 50.5 | 49.8 | 51.1 | 51.6 |
| **+Method(1)** | 160+2 | 12.6% | 50.6 | 50.0 | 50.8 | 52.0 |
| **+Method(1)&(2)** | 56+0.8 | 4.4% | 50.5 | 49.7 | 51.0 | 51.9 |

## H  CASE STUDY ON TRAINING DATA

To evaluate the training objective of our FRAME-VOYAGER, we examine random samples from the NextQA dataset after processing it through our data collection pipeline in Section 3.1. Figure 8 shows these sample cases, revealing a clear pattern: question-answer pairs with lower loss values show stronger alignment with the visual content. This correlation demonstrates that our loss-based ranking effectively identifies the most contextually relevant frames for answering questions. However, in cases with higher loss values, while the frames may contain relevant objects, they often lack sufficient visual information to fully answer the given questions.

## I  CASE STUDY ON FRAME COMBINATIONS BY FRAME-VOYAGER

To address RQ6, we thoroughly examine the findings presented in Figure 9. In Figure 10 and 11, we present additional case studies to highlight the effectiveness of our model.

In Figure 10, the uniform sampling fails to capture key objects mentioned in the query. Similarly, the CLIP-based method struggles due to its limited OCR capabilities on special fonts, often match objects like "blue food dye". In contrast, our FRAME-VOYAGER, accurately identifies the used ingredients in this video, providing sufficient video context to answer the query about which ingredients are not used.

Figure 11 further demonstrates the limitations of uniform sampling, which produces a lot of irrelevant background frames. While the CLIP-based method focuses on isolated keywords like "basketball" and "boy", it lacks the ability to connect frames meaningfully. Our method, however, selects representative frames based on the query, illustrating key events in temporal order—from the boy's training to the basketball match, and finally, the podium.

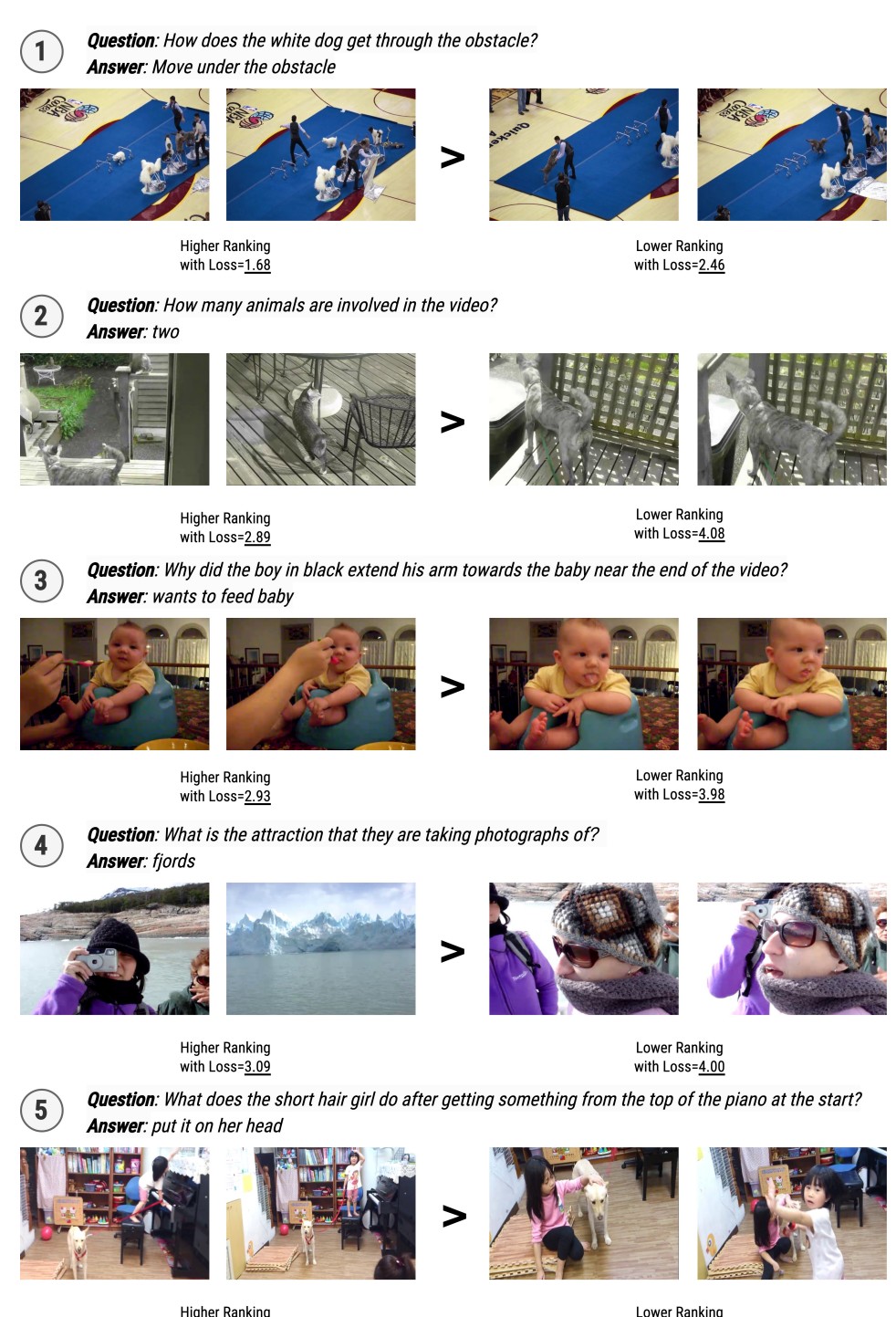

Figure 8: Case study from processed NextQA training data: These random samples show that lower loss values correlate with better visual-question alignment. The loss signal can enable learning to effectively select frame combinations for answering questions.

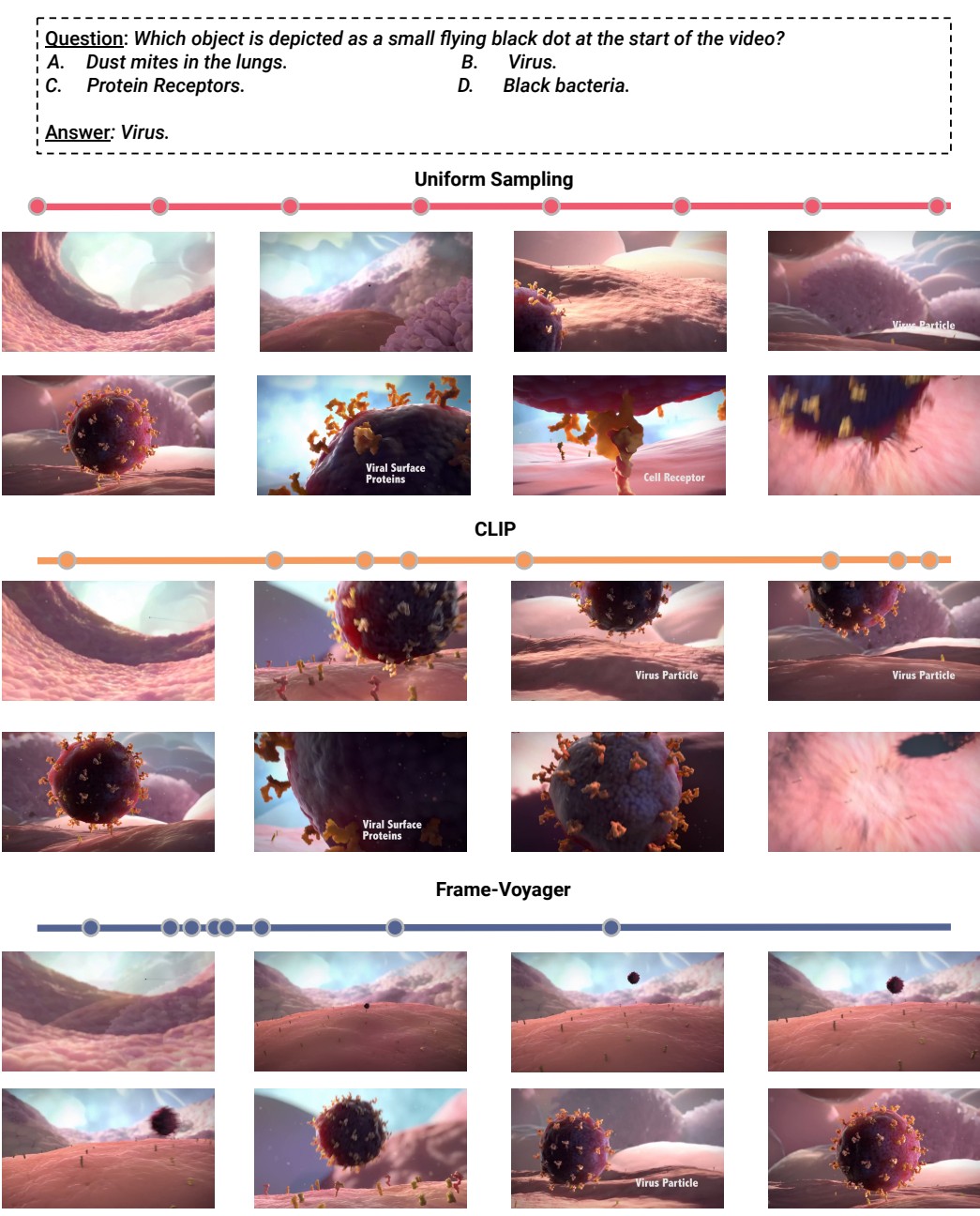

Figure 9: Case study from Video-MME: The horizontal lines represent the timeline, with points marking the time positions of frames extracted by different methods. Uniform sampling captures only a limited number of frames relevant to the query. While CLIP extracts more relevant frames, it struggles to capture the temporal dynamics of gradual zoom-in transitions. In contrast, FRAME-VOYAGER effectively selects a combination of frames that are both highly relevant to the query and accurately reflect the correct temporal sequence.

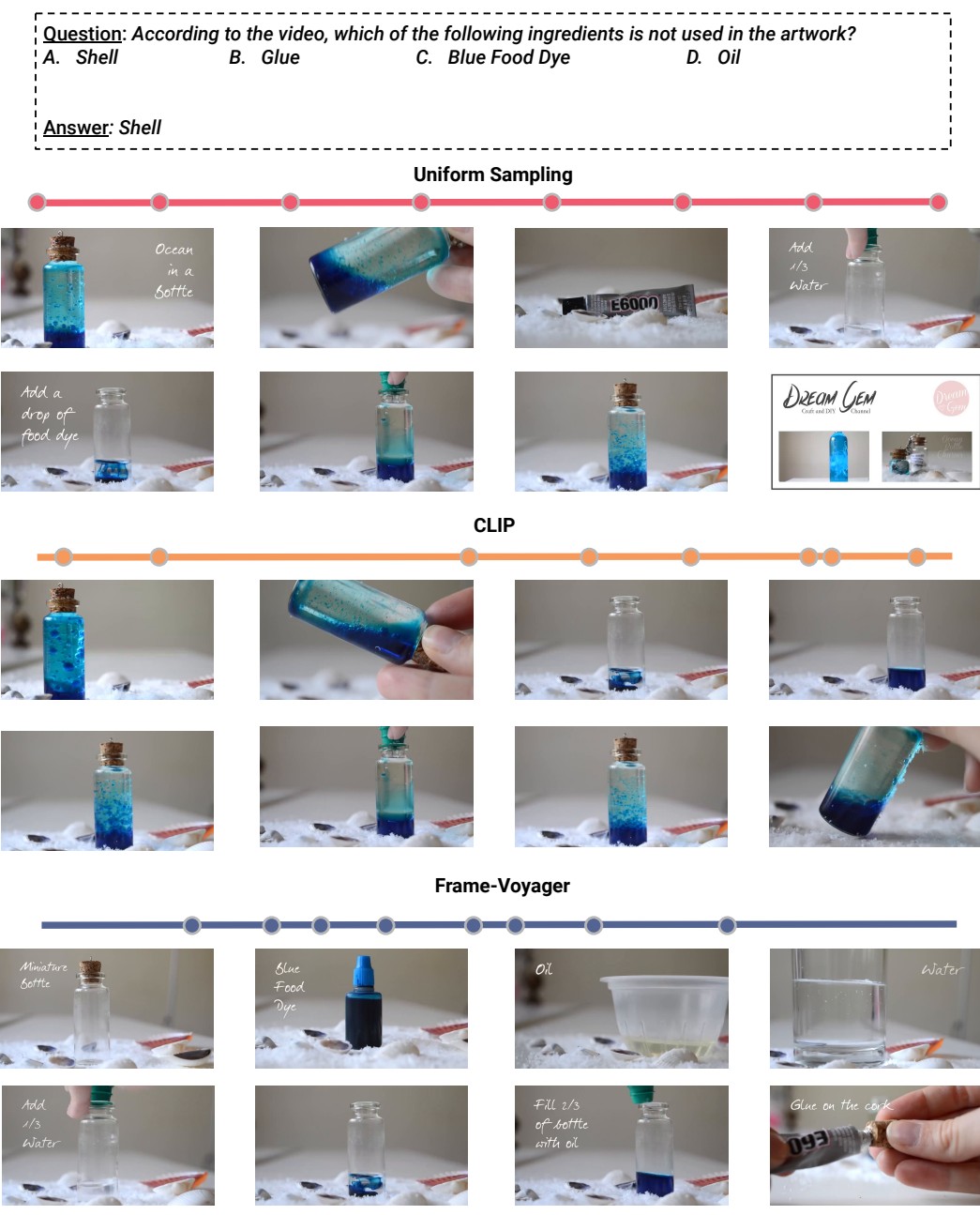

Figure 10: Case study from Video-MME: The horizontal lines represent the timeline, with points marking the time positions of frames extracted by different methods. We can see that the uniform sampling fails to capture key objects relevant to the query, while the CLIP-based method, due to its limited OCR capabilities on special fonts, incorrectly matches terms like "blue food dye". In contrast, FRAME-VOYAGER effectively identifies the used ingredients, providing the necessary context to accurately answer the query about which ingredient is not used.

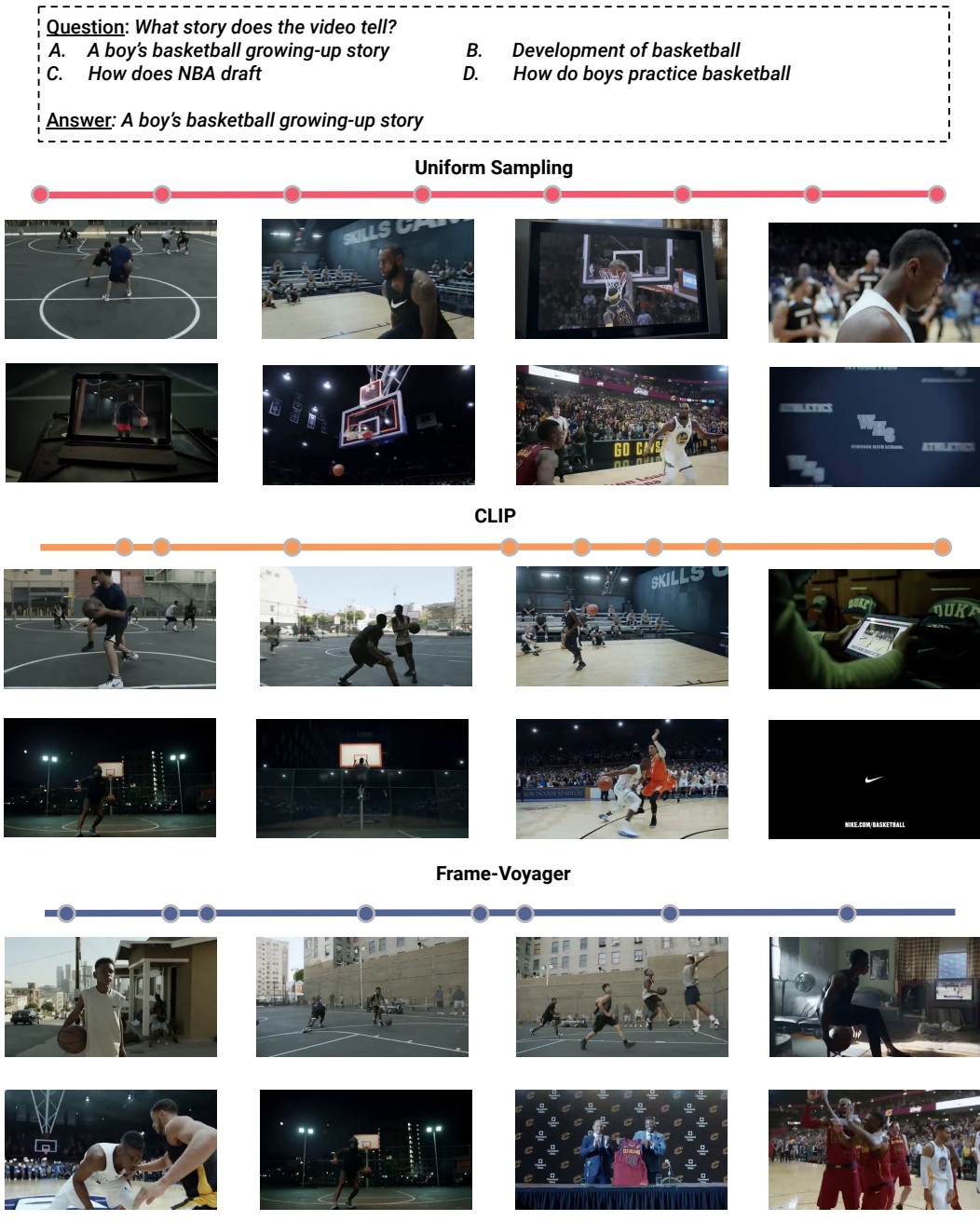

Figure 11: Case study from Video-MME: The horizontal lines represent the timeline, with points marking the time positions of frames extracted by different methods. It shows that uniform sampling produces a sequence containing many irrelevant background frames. The CLIP-based method, though focused on keywords like "basketball" and "boy", fails to capture the temporal relationships between frames. Our approach can select high-quality frames that align with the query, illustrating key events in temporal order—from "boy's training" to the "basketball match" and finally the "podium".

