# OpenReview forum: "Frame-Voyager: Learning to Query Frames for Video Large Language Models"
_ICLR.cc/2025/Conference — ICLR 2025 Poster_

### Official Review · Reviewer_ULTS · 2024-10-25

**Soundness:** 3
**Presentation:** 3
**Contribution:** 3
**Rating:** 6
**Confidence:** 5

**Summary:**

The paper presents FRAME-VOYAGER, a novel approach for improving Video Large Language Models by selecting informative frame combinations based on textual queries. Traditional methods like uniform frame sampling and text-frame retrieval do not account for variations in information density or complex instructions, leading to sub-optimal performance. FRAME-VOYAGER addresses this by learning to query frame combinations with lower prediction losses, using a pre-trained Video-LLM for supervision. The method is demonstrated to enhance performance in video question answering tasks across various benchmarks. The approach also includes a new data collection and labeling pipeline, which ranks frame combinations to train the model efficiently. FRAME-VOYAGER is proposed as a plug-and-play solution, improving Video-LLMs without significant computational overhead. Experiment results show that FRAME-VOYAGER achieves significant performance in several VQA datasets.

**Strengths:**

1.The authors designed an efficient keyframe data construction method, exploring keyframe selection using combinations of all frames. This approach achieved significant improvements across multiple datasets with 12K high-quality clips.

2.The authors state that they will open-source the data and related code, which is valuable for the open-source community.

3.The proposed method demonstrate strong performance across multiple datasets, proving the robustness of the approach, especially in Anet dataset. As a plug-and-play method, Frame-Voyager can serve as a valuable tool for video understanding.

**Weaknesses:**

1.Data scaling limitations. As the authors mentioned, selecting 4 frames from 32 results in 35960 candidates, leading to prohibitively high costs for data scaling. Furthermore, keyframe selection is even more critical for longer videos, and the costs become unmanageable when dealing with selections from 128 or 256 frames.

2.When testing on short video benchmarks, did VILA and VILA+FRAME-VOYAGER evaluate performance by selecting 2 frames out of 16? If so, please provide the results of VILA using 16 frames on the Next-QA and ANQA datasets. Additionally, I believe that comparing results based on only 2 frames is not meaningful, as most models typically evaluate performance using 8 or 16 frames.

3.Please provide additional results of the model on more benchmarks, such as MVbench and egoSchema.

**Questions:**

See the above weakness. I am interested in how the authors consider data scaling. Constructing data with only 16 or 32 frames is not very helpful for understanding long videos.

---

> ### Author Response · Authors · 2024-11-21
> **Response to the Reviewer ULTS (First Round-1)**
>
> > **W1&Q1: data scaling for longer videos**
>
> **Response:**
>
> Frame-Voyager demonstrates strong generalization capabilities to longer videos through two key aspects:
>
> (1) **First, Frame-Voyager inherently possesses generalization capabilities, as the essence of video understanding is transferable.** It is aligned with most of the existing Video-LLMs’ practices, where models are trained primarily on short videos, yet can also generalize to long videos during testing, such as VILA and LLaVA-One-Vision [1][2].
>
> For example, considering temporal reasoning tasks where the question requires locating the beginning of a video (case study shown in Fig. 6, Lines 445-485), the Frame-Voyager consistently focuses on the video openings regardless of total duration. Similarly, for content identification tasks, e.g., finding ingredients as shown in Fig. 9 (Lines 922-68), it selects relevant frames containing ingredients independent of video length.
>
> In the main paper, we sample 4 frames from 32 frames (or 2 frames from 16 frames) in training, and we directly choose 8 frames from 128 frames in inference. To further demonstrate the generalization capabilities of Frame-Voyager, we augment experiment on LLaVA-One-Vision (please kindly refer to ```General Response of First Round Rebuttal (2/4)```  for the details and the experimental results). **The results show that Frame-Voyager shows consistently better performance when varying the number of frames from 2 to 64.**
>
> (2) **Second, our data collection process can be further optimized.** We implement two optimization approaches (please kindly refer to ```General Response of First Round Rebuttal (3/4)``` for details and experimental results). The first leverages the frame-to-frame similarity and temporal proximity to filter out redundant combinations, and the second one (suggested by the Reviewer ```qbz8```) uses the CLIP to reduce search space. **The two optimizations significantly reduce the Frame-Voyager’s data construction time to only 4.4% of the original duration, while maintaining comparable performance across all benchmarks.**
>
> In summary, the elaborated two key aspects demonstrate Frame-Voyager's scalability for long video analysis through both its inherent design and optimized data collection pipeline.
>
> [1] VILA: On Pre-training for Visual Language Models. CVPR 2024
>
> [2] ​​LLaVA-OneVision: Easy Visual Task Transfer. ArXiv 2024.

---

> ### Author Response · Authors · 2024-11-21
> **Response to the Reviewer ULTS (First Round-2)**
>
> > **W2: 8 or 16 frames for short video benchmarks**
>
> **Response:**
>
> We appreciate the reviewer's suggestion regarding the short video evaluation. We have expanded our experiments to include 8 and 16-frame evaluations on NextQA and ANQA datasets. The results are shown below.
>
> |                    | ANQA w/ 8 frames | NextQA w/ 8 frames | ANQA w/ 16 frames | NextQA w/ 16 frames |
> |--------------------|------------------|--------------------|-------------------|---------------------|
> | VILA-8B            | 53.7             | 55.6               | 56.2              | 65.7                |
> | **+Frame-Voyager** | 55.7             | 60.8               | 56.2              | 65.7                |
> | VILA-40B           | 56.8             | 62.9               | 57.5              | 71.1                |
> | **+Frame-Voyager** | 57.9             | 67.3               | 57.5              | 71.1                |
>
> Our results show Frame-Voyager consistently improves performance with 8-frame processing across both model sizes and datasets. Note that with 16 frames, Frame-Voyager's behavior matches uniform sampling as no frame selection is required.
>
> We will add these expanded results in our revised paper to provide a more comprehensive evaluation of our method's effectiveness across different frame settings.
>
> ***
>
> > **W3: Addition results on more benchmarks**
>
> **Response:**
>
> We are currently conducting the relevant experiments and will provide the corresponding results very soon!
> ***

---

> ### Author Response · Authors · 2024-11-23
> **Response to the Reviewer ULTS (First Round-3)**
>
> > **W3: Addition results on more benchmarks**
>
> **Response:**
>
> Thanks for the waiting, we provide experimental results on MVbench in the table below. The experiments are conducted with 8 frames as suggested by Weakness # 2 of Reviewer ```ULTS```.
>
> |                    | **MVBench (Avg. Video Length=13s)** |
> |--------------------|-------------------------------------|
> | VILA-8B            | 40.1                                |
> | **+Frame-Voyager** | 41.1                                |
> | VILA-40B           | 56.2                                |
> | **+Frame-Voyager** | 57.3                                |
>
> ***
>
> \
> \
> \
> Please feel free to let us know if you have any additional questions. We would be happy to address them.

---

> > ### Author Response · Authors · 2024-11-25
> > **Kind Reminder: Seeking Reviewer Feedback for Author/Reviewer Discussion Phase**
> >
> > Dear Reviewer ULTS,
> >
> > We hope this message finds you well. We would like to express my gratitude for the time and effort that you have dedicated to the review process of our submission.
> >
> > Furthermore, we are writing to kindly remind you that the rebuttal period is coming to a close. Your feedback and evaluation are crucial to the progress of our work, and we value your expertise and insights immensely.
> >
> > During rebuttal, we have thoroughly addressed all the queries and concerns you raised regarding our submission. Each point has been carefully considered, and we have provided detailed responses.
> >
> > We are eager to engage in further discussions with you on the responses and any other aspects of our study. Your feedback plays a pivotal role in shaping the direction of our research, and we are keen to hear your thoughts and suggestions.
> >
> > Thank you for your attention to this matter.
> >
> > Best regards, Authors of submission #13344

---

> > > ### Comment · Reviewer_ULTS · 2024-11-25
> > > **Replying to Author**
> > >
> > > The authors have partly addressed my concerns, the method in **General Response of First Round Rebuttal (3/4)** appears to be highly efficient and maintains comparable performance, significantly reducing the costs associated with data scaling. And I still have some concerns:
> > >
> > > 1.The training data used actually includes the training sets of Next-QA and ANQA. Therefore, I am uncertain whether the performance improvement on these two datasets is due to the proposed method's effectiveness or simply because in-domain datasets were used during training.
> > >
> > >
> > > 2.The results in First Round-2 show a 10% difference between the 8-frame and 16-frame results on Next-QA, which seems unreasonable. How can this issue be explained?
> > >
> > >
> > > 3.Since mvbench also includes data from Next-QA and ANQA, perhaps the authors could present results on additional datasets (EgoSchema, STAR, etc.) to provide a more comprehensive evaluation.

---

> > > > ### Author Response · Authors · 2024-11-25
> > > > **Response to the Reviewer ULTS (Second Round-1)**
> > > >
> > > > Thank you for your follow-up questions and comments. We truly appreciate the time and effort you’ve taken to engage with our work in such detail. Your feedback has been very valuable in helping us improve the manuscript.
> > > >
> > > > Below, we provide detailed responses to your follow-up queries:
> > > >
> > > > ***
> > > >
> > > > > **Q1: Benefit from in-domain dataset**
> > > >
> > > > **Response:**
> > > >
> > > > First, as outlined in Lines 294-295, our selection of VideoChatGPT and NextQA is motivated by their question diversity. While VideoChatGPT [1] may share some raw video content with ActivityNetQA [2] (ANQA), the question-answer pairs in VideoChatGPT are entirely new, as stated in their abstract: "we introduce a new dataset of 100,000 video-instruction pairs used to train Video-ChatGPT." This indicates a distinct data distribution between VideoChatGPT (our training dataset) and ActivityNet QA.
> > > >
> > > > Second, it is important to note that Video Question Answering performance is fundamentally determined by the inference model—in our case, the frozen VILA. Frame-Voyager's architecture precludes direct intervention in the final results, thereby inherently limiting potential shortcut effects caused by the in-domain data.
> > > >
> > > > Third, our experimental validation encompasses multiple "zero-overlapped" benchmarks. The main paper presents results on two widely adopted benchmarks: Video-MME and MLVU. During the rebuttal phase, we extend our evaluation to include two additional benchmarks: MVBench and STAR. Notably, we have verified that MVBench has no overlap with ANQA and NextQA (detailed clarification provided in the response below).
> > > >
> > > > In conclusion, our experimental evidence, derived from four entirely independent benchmarks, demonstrates the effectiveness of Frame-Voyager on out-of-distribution benchmarks while exhibiting minimal susceptibility to in-domain data influence.
> > > >
> > > > [1] Video-ChatGPT: Towards Detailed Video Understanding via Large Vision and Language Models, ACL 2024.
> > > >
> > > > [2] ActivityNet-QA: A Dataset for Understanding Complex Web Videos via Question Answering, AAAI 2019.
> > > >
> > > > ***
> > > >
> > > > > **Q2: Results on Next-QA**
> > > >
> > > > **Response:**
> > > >
> > > > We have thoroughly validated our implementation of NextQA and confirmed its correctness. The code will be publicly released to ensure reproducibility.
> > > >
> > > > Meanwhile, we conduct analysis on the cases which are predicted correctly due to the higher number of frames.
> > > > * The distribution of question types in the overall NextQA test set is shown in row 1 of Table 1.
> > > > * The second row shows the percentage of different question types in the samples that are answered correctly due to the higher number of frames. For example, by increasing the number of frames from 8 to 16, there are additional 10 samples being predicted correctly, and there are only 1 sample asking why within the 10 samples, thus the percentage of “why” is 10%.
> > > >
> > > > From the results, we can see that increasing the number of frames will be more helpful for “why” questions and temporal reasoning problems asking about “before & after”. It demonstrates that increasing frame sampling particularly enhances performance on questions requiring causal and temporal reasoning.
> > > >
> > > >
> > > > |                      | Why   | Before & After | How   | When  | Count | Location | Other |
> > > > |----------------------|-------|---------|-------|-------|-------|----------|-------|
> > > > | Overall Distribution       | 38.9% | 17.4%   | 13.7% | 13.6% | 3.8%  | 5.6%     | 7.0%  |
> > > > | Correct Sample Ratio Benefiting from 8-frame to 16-frame | 48.0% | 19.1%   | 11.6% | 13.2% | 1.6%  | 2.4%     | 4.2%  |

---

> > > > > ### Author Response · Authors · 2024-11-25
> > > > > **Response to the Reviewer ULTS (Second Round-2)**
> > > > >
> > > > > > **Q3: Results on more benchmarks**
> > > > >
> > > > > **Response:**
> > > > >
> > > > > First, we wish to clarify that MVBench contains no data overlap with either NextQA or ActivityNetQA. MVBench paper [1] introduces both an evaluation benchmark (MVBench) and a Video MLLM model (VideoChat2 Model). Table 1 in [1] explicitly delineates the video sources utilized in MVBench, where NextQA and ActivityNetQA are not included. We think that the Reviewer ULTS may have been inadvertently misled by Figure 3 in [1]. Figure 3 illustrates the instruction-tuning data for the VideoChat2 Model, where VideoChatGPT and NextQA are indeed included.
> > > > >
> > > > > Second, in response to Reviewer ULTS's request, we conduct additional experiments on Egoshema and STAR. The results are presented below.
> > > > >
> > > > >
> > > > > |                    | **STAR** | **Egoschema** |
> > > > > |--------------------|----------|---------------|
> > > > > | VILA-8B            | 48.3     | 53.3          |
> > > > > | **+Frame-Voyager** | 50.5     | 53.6          |
> > > > > | VILA-40B           | 62.1     | 63.2          |
> > > > > | **+Frame-Voyager** | 63.8     | 63.3          |
> > > > >
> > > > > Our results reveal that Frame-Voyager performs well on STAR, e.g., +2.2% on VILA-8B and +1.7% on VILA-40B.
> > > > >
> > > > > |                    | **Egoschema** |
> > > > > |--------------------|---------------|
> > > > > | VILA-8B            | 53.3          |
> > > > > | +CLIP              | 52.0          |
> > > > > | +SigLIP            | 52.4          |
> > > > > | +VILA Embedding    | 53.4          |
> > > > > | **+Frame-Voyager** | **53.6**          |
> > > > >
> > > > > There is marginal improvement on the Egoschema benchmark [2] with Frame-Voyager. Results with other frame selection methods either fail to improve performance or lead to decreased performance on the Egoschema dataset.
> > > > >
> > > > > Upon detailed examination of the dataset, we identify the main reason: ambiguous pronoun and camera wearer (cameraman) related question. In our random sampling of 50 instances, 44 instances contain the character "c" rather than the conventional pronoun to represent the human. For example, “what was the primary tool used by c in the video” and “what can be inferred about c's assessment of the plants during the video?”. From the original paper of Egoschema [2], we find that “c” represents the “camera wearer”, which may not even appear in the video.  Such special characteristics impede Frame-Voyager's ability to identify truly query-relevant frame combinations.
> > > > >
> > > > > [1] MVBench: A Comprehensive Multi-modal Video Understanding Benchmark, CVPR 2024.
> > > > >
> > > > > [2] EgoSchema: A Diagnostic Benchmark for Very Long-form Video Language Understanding, NeurIPS 2023.
> > > > >
> > > > > ***
> > > > >
> > > > > \
> > > > > \
> > > > > \
> > > > > We hope these responses address your concerns. Please let us know if there are any additional questions.

---

> > > > > > ### Author Response · Authors · 2024-11-30
> > > > > >
> > > > > > Dear Reviewer ULTS,
> > > > > >
> > > > > > We hope this message finds you well. Thank you again for your thoughtful feedback and the time you have already dedicated to reviewing our submission. Your insights have been instrumental in improving our work, and we greatly appreciate your previous responses.
> > > > > >
> > > > > > As the rebuttal period is nearing its end, we want to kindly follow up on our latest responses to your comments. We have carefully addressed the points you raised and would value any further feedback or suggestions you might have.
> > > > > >
> > > > > > Your input is crucial to the progress of our work, and we are eager to engage in any additional discussions needed to ensure all your concerns are addressed.
> > > > > >
> > > > > > Thank you once again for your time and support. We look forward to hearing from you soon.
> > > > > >
> > > > > > Best regards, Authors of submission #13344

---

### Official Review · Reviewer_qbz8 · 2024-10-28

**Soundness:** 1
**Presentation:** 3
**Contribution:** 2
**Rating:** 6
**Confidence:** 4

**Summary:**

This paper introduces Frame-Voyager, a novel frame selection method designed to improve the performance of Video-LLMs on video question-answering (QA) tasks. The key contributions are (1) an automated data collection strategy that ranks frame combinations according to the QA loss from a reference Video-LLM, and (2) a frame selection module integrated with a Video-LLM, trained using this ranked data. When applied to VILA, Frame-Voyager outperforms the uniform sampling baseline across four video QA benchmarks.

**Strengths:**

**1. Originality:** The novel formulation of frame selection as a ranking problem is compelling. This approach minimizes the need for extensive manual labeling to obtain ground truth, offering a fresh and efficient perspective on the task.

**2. Significance:** Frame selection is a critical challenge in video understanding due to the high dimensionality of video data and the computational demands involved. Traditional uniform sampling often misses relevant content, making this problem a key focus for advancing the field.

**3. Clarity:** The paper is well-written, with a clear explanation of the motivation and methodology. The authors effectively communicate the concepts, making the approach easy to understand.

**Weaknesses:**

**1. Data Collection Cost:**
The number of frame combinations increases exponentially with M and T (L166), leading the authors to limit M to 16 or 32 and T to 2 or 4. Despite these restrictions, evaluating $C(32, 4) \approx 36K$ combinations for a single QA sample is still costly, raising concerns about data collection efficiency. Thus, this paper needs to provide more details on the computational resources and time required for data collection.

**2. Generalization to More Frames:**
The method is trained to select up to T=4 frames from M=32 candidates. Although the authors claim the model generalizes well to larger M and T values (L170), Fig. 3 shows diminishing gains over uniform sampling while increasing the number of frames. Despite the authors attributing this to benchmark constraints (L431), prior research (e.g., LongVILA [1], LongVA [2]) has shown that additional input frames can enhance QA performance, suggesting potential limits in generalization. Given the impracticality of increasing M and T for data collection, robust generalization is crucial. Thus, the authors should extend their experiments to more input frames, possibly integrating Frame-Voyager with LongVILA, to help clarify the generalization capabilities and limitations of Frame-Voyager.

**3. Fairness of Comparisons:**
The comparison between Frame-Voyager and CLIP is not fully equitable due to three factors: (i) Frame-Voyager uses superior backbones (SigLIP or InternViT-6B); (ii) it is trained on specialized ranking data, unlike CLIP; and (iii) it has additional parameters like self-attention modules. Since (ii) and (iii) are core contributions of Frame-Voyager, a fairer comparison would involve using SigLIP or InternViT-6B models with and without training on the collected data. Additionally, the authors should specify the versions of CLIP and SigLIP used.

**4. Latency Analysis:**
The authors claim that Frame-Voyager outperforms LongVILA while processing fewer frames, but this only holds for the "Long" subset of the Video-MME benchmark. In other scenarios, LongVILA performs better. Given the extra parameters introduced by Frame-Voyager, a detailed efficiency comparison between VILA+Frame-Voyager and LongVILA is necessary, including latency, memory usage, and computational complexity.

---
[1] Xue F, Chen Y, Li D, et al. Longvila: Scaling long-context visual language models for long videos[J]. arXiv preprint arXiv:2408.10188, 2024.

[2] Zhang P, Zhang K, Li B, et al. Long context transfer from language to vision[J]. arXiv preprint arXiv:2406.16852, 2024.

**Questions:**

My questions focus primarily on the weaknesses outlined above:

1. What are the data collection costs, particularly for the VideoChatGPT data?

2. How well does the method generalize when increasing the number of frame candidates and selected frames?

3. Can the proposed model outperform SigLIP and InternViT-6B, especially when they are fine-tuned on the same collected data?

4. What is the latency of the method?

5. A suggestion: Could CLIP-like models enhance data collection efficiency? Specifically, using them to estimate query-frame similarities might identify frames with extremely high or low similarities, indicating positive or negative frame combinations. This could reduce the search space and improve data collection efficiency.

---

> ### Author Response · Authors · 2024-11-21
> **Response to the Reviewer qbz8 (First Round-1)**
>
> > **W1.1&Q1: Cost and detail for data collection**
>
>
> **Response:**
>
> For the data collection, our choice of smaller frame combinations (M=16/32, T=2/4) aligns with current practice. The leading Video-LLMs, such as VILA and LLaVA-One-Vision, are primarily trained on short video segments [1, 2], but can still generalize to long videos during testing. Specifically, generating frame combinations based on C(32, 4) for the VideoChatGPT (100k) dataset requires 32 A100 GPUs over approximately 40 hours. For the NextQA (35k) dataset, using C(16, 2), it takes 8 A100 GPUs about 1 hour.
>
> [1] VILA: On Pre-training for Visual Language Models. CVPR 2024
>
> [2] ​​LLaVA-OneVision: Easy Visual Task Transfer. ArXiv 2024.
>
> ***
>
> > **W1.2&Q5: Data collection efficiency**
>
>
> **Response:**
>
> Thanks for the valuable suggestions. First, we would like to highlight that our data collection process is **automated and one-off**, allowing the resulting data (generated by VILA-8B) to be **directly used** by other Video LLMs, i.e., results of VILA-40B in Table 1 (Lines 292-293) and results of LLaVA-One-Vision-7B in the ```General Response of First Round Rebuttal (1/4)```. We will open-source our algorithm and also the generated dataset after the paper's acceptance.
>
> Furthermore, to address the efficiency concern, we implement two optimization approaches (please kindly refer to ```General Response of First Round Rebuttal (3/4)``` for details and experimental results). The first leverages the frame-to-frame similarity and temporal proximity to filter out redundant combinations, and the second one (suggested by the Reviewer ```qbz8```) uses the CLIP to reduce search space. **The two optimizations significantly reduce the Frame-Voyager’s data construction time to only 4.4% of the original duration, while maintaining comparable performance across all benchmarks.**
>
> ***
>
> > **W2.1 More input frames can enhance QA performance?**
>
>
> **Response:**
>
> We respectfully disagree with the claim regarding the benefits of additional input frames for QA performance.
>
> We agree that arbitrarily increasing the token length of the model may not always lead to improvement [1] (Lines 038-040). This observation is also aligned with the LongVA-7B's results on the Video-MME benchmark (Table 4 in LongVA paper [2]), where performance plateaus or degrades with more frames.
>
> For instance, on short videos, the LongVA-7B’s accuracy peaks at 61.1% with 32 frames, marginally improves to 61.4% with 64 frames, but drops back to 61.1% at 128 frames. Similarly, the LongVA-7B’s overall performance shows minimal gains from 52.4% (64 frames) to 52.6% (128 frames), and even declines to 51.8% with 384 frames.
>
> [1] Lost in the middle: How language models use long contexts. TACL 2024.
>
> [2] Long Context Transfer from Language to Vision. ArXiv 2024.
>
>
> ***
>
> > **W2.2&Q2 Generalization ability of Frame-Voyager**
>
> **Response:**
>
> Thanks for raising this important point. Please kindly refer to ```General Response of First Round Rebuttal (2/4)``` for the comprehensive analysis and experimental results regarding the generalization of Frame-Voyager on higher numbers of frames. It demonstrates that Frame-Voyager maintains superior performance across frame numbers ranging from 2 to 64.
>
> Additionally, with the optimized data collection pipeline mentioned in ```General Response of First Round Rebuttal (3/4)``` and the response to W1.2&Q5 above, we believe that the generalization ability of Frame-Voyager could be further enhanced through increasing the numbers of frames in data collection.

---

> > ### Author Response · Authors · 2024-11-21
> > **Response to the Reviewer qbz8 (First Round-2)**
> >
> > > **W3.1&Q3: Fairness of comparison**
> >
> > **Response:**
> >
> > We would like to clarify several points about our comparison methodology.
> > * First, regarding the implementation of CLIP in our paper: we deploy a separate CLIP model specifically for selecting frames. The CLIP model contains 336M parameters and requires standalone deployment. In contrast, our Frame-Voyager is a lightweight plug-in module that adds only ~20M parameters.
> > * Second, Frame-Voyager does not include SigLIP or InternViT-6B. The SigLIP or InternViT-6B are the built-in visual encoders for VILA-8B and VILA-40B, respectively.
> >
> > To further validate the effectiveness of our approach, which benefits from the frame combination design, we list two groups of experiment below:
> >
> > * We already include an experiment setting by replacing the CLIP with VILA Embedding (Table 2, Line 336) for frame selection. VILA Embedding utilizes the visual feature generated by the SigLIP & projector and the textual feature from word embedding in LLM. The implementation details are included in Appendix A (Lines 821-824). **The result of VILA Embedding is 48.8 while Frame-Voyager achieves 50.5.**
> > * As suggested by the Reviewer ```qbz8```, we also replace the CLIP model used in the CLIP-based frame selection approach with SigLIP and InternViT-6B for ablation studies. **The results of SigLIP and InternViT-6B are 48.3 and 49.1, respectively, comparable to CLIP's performance of 48.5.**
> > **The results from both two experimental groups demonstrate that the effectiveness of our approach is not derived from the superior backbones.** We will also include these extra baselines in the revised version.
> >
> > Finally, we would like to highlight that none of these baseline modules (CLIP, SigLIP, InternViT-6B) can be trained on frame combination data, as they're limited to single image-text pairs. Frame-Voyager's advantage lies in its ability to model relationships between multiple frames and questions while learning global features simultaneously.
> >
> > ***
> >
> > > **W3.2 Versions of CLIP and SigLIP**
> >
> > **Response:**
> >
> > The CLIP model used is clip-vit-large-patch14-336 [Link: https://huggingface.co/openai/clip-vit-large-patch14-336], while SigLIP model used in VILA-8B and above experiments is google/siglip-so400m-patch14-384 [Link: https://huggingface.co/google/siglip-so400m-patch14-384]. We will add these details in the Implementation Details section of our paper.
> >
> > ***

---

> > ### Comment · Reviewer_qbz8 · 2024-11-26
> >
> > Thank you for your detailed rebuttal. I appreciate the authors' efforts in addressing my feedback, particularly in optimizing data collection efficiency.
> >
> > However, I remain concerned regarding **W2 (Generalization to More Frames)**. While the authors argue that more frames do not always lead to improved performance, this contradicts the findings of LongVILA, which I referenced in my previous review. The authors did not provide a satisfactory explanation for this discrepancy. Additionally, while the experiments demonstrate that Frame-Voyager generalizes well from 4 to 32 frames, real-world tasks may require more varied numbers of frames to effectively support reasoning. I am still unconvinced by the proposed method's generalizability and flexibility, as also noted by Reviewer `6u8S`.

---

> ### Author Response · Authors · 2024-11-21
> **Response to the Reviewer qbz8 (First Round-3)**
>
> > **W4: Efficiency comparison between Frame-Voyager and LongVILA**
>
> **Response:**
>
> We appreciate the reviewer's concern regarding efficiency comparisons.
> * First, we would like to clarify that Frame-Voyager and LongVILA differ fundamentally in their approaches: LongVILA requires complete model re-training with five stages, while Frame-Voyager functions as a modular enhancement to the existing VILA architectures.
> * Second, to facilitate a thorough comparison, we present detailed efficiency comparison below.
>
> |                   | **Plug-and-Play** | **Training Hours** | **Params v.s. Vanilla VILA** | **Latency** | **Memory** |
> |-------------------|-------------------|--------------------|----------------------------|-------------|------------|
> | LongVILA      | No                | 5,400+ GPU hours    |  +0%                          | 8.654s      | 74,293 MiB |
> | Frame-Voyager  | Yes               | ~64 GPU hours       | +0.2%                       | 1.696s      | 23,069 MiB |
> The table shows several key advantages of Frame-Voyager:
>   * Resource Efficiency: Frame-Voyager requires only ~64 GPU hours of training compared to LongVILA's estimated 5,400+ hours, making it significantly more accessible and cost-effective.
>   * Minimal Parameter Increase: Frame-Voyager adds only 20M parameters (0.2% increase) to the base VILA model.
>   * Inference Latency:  Frame-Voyager demonstrates superior computational efficiency with a 5x reduction in latency (1.696s vs 8.654s) and a 3x reduction in memory consumption (23,069 MiB vs 74,293 MiB) compared to LongVILA.
>
> These results highlight Frame-Voyager's practical advantages in terms of deployment flexibility, resource utilization, and computational efficiency, while maintaining competitive performance on video understanding tasks.
>
> ***
>
> > **Q4: Latency of the method**
>
> **Response:**
>
>
> Thanks for the valuable suggestions. Please kindly refer to the ```General Response of First Round Rebuttal (4/4)``` for the analysis of overall efficiency.
>
> In summary, the Frame-Voyager only introduces a tiny computational overhead. The minimal increases in parameter count and memory usage suggest that our method scales efficiently while achieving superior video understanding capabilities.
>
> ***
> \
> \
> \
> Please kindly let us know if you have any additional follow-up questions, and we would be happy to address them.

---

> ### Author Response · Authors · 2024-11-23
> **Kind Reminder: Seeking Reviewer Feedback for Author/Reviewer Discussion Phase**
>
> Dear Reviewer qbz8,
>
> We hope this message finds you well. We would like to express my gratitude for the time and effort that you have dedicated to the review process of our submission.
>
> Furthermore, we are writing to kindly remind you that the rebuttal period is coming to a close. Your feedback and evaluation are crucial to the progress of our work, and we value your expertise and insights immensely.
>
> During rebuttal, we have thoroughly addressed all the queries and concerns you raised regarding our submission. Each point has been carefully considered, and we have provided detailed responses.
>
> We are eager to engage in further discussions with you on the responses and any other aspects of our study. Your feedback plays a pivotal role in shaping the direction of our research, and we are keen to hear your thoughts and suggestions.
>
> Thank you for your attention to this matter.
>
> Best regards, Authors of submission #13344

---

> > ### Author Response · Authors · 2024-11-25
> > **Kind Reminder: Seeking Reviewer Feedback for Author/Reviewer Discussion Phase**
> >
> > Dear Reviewer qbz8,
> >
> > We regret any disturbance this message may cause and appreciate your understanding.
> >
> > We have posted our responses to address your concerns.
> >
> > With **less than two days** remaining in the discussion period, we kindly hope we can get your feedback.
> >
> > We understand that this is quite a busy period, but we’d appreciate it very much if you could take some time to read our responses. If there are other comments or concerns, we will be happy to address them during the response period.
> >
> > Regards,
> >
> > Author(s) of submission #13344

---

> ### Author Response · Authors · 2024-11-28
> **Response to the Reviewer qbz8 (Second Round-1)**
>
> Thank you for your follow-up questions and comments. We truly appreciate the time and effort you’ve taken to engage with our work in such detail. Your feedback has been very valuable in helping us improve the manuscript.
>
> Below, we provide detailed responses to your follow-up queries:
>
> ***
>
> ***Q1: Marginal effect of more frames***
>
> In the Reviewer ```qbz8```’s original review, it is said that:
>
> >Reviewer qbz8: prior research (e.g., LongVILA [1], LongVA [2]) has shown that additional input frames can enhance QA performance.
>
> In the Reviewer ```qbz8```’s latest review, it is mentioned that:
>
> >Reviewer qbz8: The authors argue that more frames do not always lead to improved performance, this contradicts the findings of LongVILA.
>
> ***
>
> We still argue that **more frames do not always lead to improved performance**.
>
> First, we delineate the results from three SOTA backbones, i.e., LongVA, LLaVA-One-Vision and LongVILA, to demonstrate that the improvement benefiting more frames has an upper bound.
>
> Second, we give the explanation from the angle of information entropy along with an intuitive example to further illustrate the diminishing marginal effect.
>
> ***
>
> **Section #1: Quantitative Results**
>
> In ```Response to the Reviewer qbz8 (First Round-1)```, **we have already cited the results from the LongVA paper [1]  which actually supports our argument, i.e., the improvement benefiting more frames has an upper bound.**
>
> < For instance, on short videos, the LongVA-7B’s accuracy peaks at 61.1% with 32 frames, marginally improves to 61.4% with 64 frames, but drops back to 61.1% at 128 frames. Similarly, the LongVA-7B’s overall performance shows minimal gains from 52.4% (64 frames) to 52.6% (128 frames), and even declines to 51.8% with 384 frames.
>
> For convenience, we copy the results from the Table 4 of LongVA paper [1] below.
>
> |**_ LongVA_**                  | **8 frames** | **16 frames** | **32 frames** | **64 frames** | **128 frames** | **384 frames** |
> |-------------------------|---------|----------|----------|----------|-----------|-----------|
> | Video-MME w/o subtitles | 47.9    | 49.7     | 51.8     | 52.4     | 52.6      | 51.8      |
>
>
> In ```General Response of First Round Rebuttal (2/4)```, adopting LLaVA-One-Vision as backbone, we have included the results on Video-MME with different numbers of frames from 2 to 128 on uniform sampling and Frame-Voyager (128 candidate frames). **The results also indicate that there are clear upper bounds. Frame-Voyager shows consistently better performance when varying the number of frames from 2 to 64.**
>
> | **_LLaVA-One-Vision-7B_** | **2 Frames** | **4 Frames** | **8 Frames** | **16 Frames** | **32 Frames** | **64 Frames** | **128 Frames** |
> |---------------------------|--------------|--------------|--------------|---------------|---------------|---------------|----------------|
> | **Uniform Sampling**      | 44.3         | 49.6         | 53.3         | 57.2          | 58.2          | 58.0          | 58.2           |
> | **Frame-Voyager**         | 51.4         | 54.8         | 57.5         | 59.2          | 60.4          | 60.5          | 58.2           |
> | **Gains**                 | +7.1         | +5.2         | +4.2         | +2.0          | +2.2          | +2.5          | Same Input     |
>
>
> In LongVILA [2], the same conclusion could be made based on Figure 3 (a). **It is clear that the gains provided by each additional frame are diminishing.** Here we copy the averaged results from Figure 3 (a) in the LongVILA paper for reference.
>
> | **_LongVILA_**              | **8-frame** | **16-frame** | **32-frame** | **64-frame** | **128-frame** | **256-frame** |
> |-----------------------------|-------------|--------------|--------------|--------------|---------------|---------------|
> | **Video-MME w/o subtitles** | 43          | 45.7         | 48.0         | 48.3         | 49.3          | 50.6          |
>
> **The results from LongVA, LLaVA-One-Vision, and LongVILA demonstrate that increasing the number of input frames yields diminishing returns, with performance improvements reaching a plateau.**
>
> References:
>
> [1] Long Context Transfer from Language to Vision, ArXiv 2024.
>
> [2] LongVILA: Scaling Long-Context Visual Language Models for Long Videos, ArXiv 2024.

---

> > ### Author Response · Authors · 2024-11-28
> > **Response to the Reviewer qbz8 (Second Round-2)**
> >
> > ***Q1: Marginal effect of more frames (continued)***
> >
> > **Section #2: Qualitative Explanation**
> >
> > For any given (video, question) pair, the amount of information required for an answer depends on both the video's information density and the query's complexity. The most direct explanation for upper bound performance (diminishing marginal effect) is that, for the same raw video, the more sampled frames there are, **the information entropy brought by the additional frames will keep decreasing**.
> >
> > We can imagine an extreme case where a raw video depicts a person sitting in front of a computer and writing a paper. The question is "how many people appeared in the video". Video-LLMs may only need one frame to answer correctly. In this case, feeding one hundred frames or one thousand frames does not generate any additional information.
> >
> > ***
> >
> > We hope that the Quantitative Results and Qualitative Explanation above could help the Reviewer ```qbz8``` to understand that continuously increasing the number of input frames will not always bring improvements, i.e., the marginal effect and additional information entropy keep decreasing.
> >
> > ***
> >
> > ***Q2: Generalization of Frame-Voyager***
> >
> > The Reviewer ```qbz8``` has recognized that our method can generalize up to 32 frames in the latest response.
> >
> > We would like to first correct the number **32**: We have already presented the results for 64 frames in ```General Response to the First Round of Review (2/4)```. It can be observed that when the number of frames varies from 2 to 64, Frame-Voyager consistently exhibits better performance.
> >
> > To further demonstrate the generalization ability of our method, we apply larger frame numbers on Frame-Voyager by increasing the number of candidate frames to 512 to explore more input frames. **From the results, we observe that Frame-Voyager can generalize to 160 Frames.**
> >
> > | **_LLaVA-One-Vision-7B_** | **96 Frames** | **128 Frames** | **160 Frames** | **192 Frames** |
> > |---------------------------|---------------|----------------|----------------|----------------|
> > | **Uniform Sampling**      | 58.1          | 58.2           | 58.2           | 58.0 |
> > | **Frame-Voyager**         | 60.6          | 60.9           | 61.0           | 60.9 |
> >
> >
> > We believe for real-world tasks, consistently feeding very long video inputs to Video-LLMs is inefficient. Our results demonstrate that we can achieve a good balance between computational efficiency and model performance with Frame-Voyager.
> >
> > ***
> >
> > ***Q3: Flexibility of Frame-Voyager***
> >
> > We have already addressed the flexibility issue in ```Response to the Reviewer 6u8S (First Round-1)``` and the results have been acknowledged by the Reviewer ```6u8S```. Frame-Voyager is able to seamlessly adapt for dynamic frame selection (without any modification).
> >
> > Specifically, we have conducted experiments on two different backbones, i.e., VILA-8B and LLaVA-One-Vision.
> >
> > | VILA-8B                        | **Num. of Frames (Avg.)** | **Video-MME (w/o subtitles)** |
> > |-----------------------------------|--------------------|---------------|
> > | Uniform Sampling              | 4                  | 44.7          |
> > | Frame-Voyager + Fixed Num.    | 4                  | 47.9          |
> > | Uniform Sampling             | 8                  | 47.5          |
> > | Frame-Voyager + Fixed Num.    | 8                  | 50.5          |
> > | **Frame-Voyager + Dynamic Num.**  |  **4.1**                  | **49.6**          |
> >
> >
> > | LLaVA-One-Vision                 | **Num. of Frames (Avg.)** | **Video-MME (w/o subtitles)** |
> > |----------------------------------|---------------------------|-------------------------------|
> > | Uniform Sampling                 | 8                         | 53.3                          |
> > | Frame-Voyager + Fixed Num.       | 8                         | 57.5                          |
> > | Uniform Sampling                 | 16                        | 57.2                          |
> > | Frame-Voyager + Fixed Num.       | 16                        | 59.2                          |
> > | Uniform Sampling                 | 32                        | 58.2                          |
> > | Frame-Voyager + Fixed Num.       | 32                        | 60.4                          |
> > | **Frame-Voyager + Dynamic Num.** | **13.3**                  | **59.1**                      |
> >
> >
> > **The results confirm Frame-Voyager's effectiveness and flexibility in dynamically selecting frames.**
> >
> > ***
> >
> > \
> > \
> > \
> > We hope these responses address your concerns. Please let us know if there are any additional questions.

---

> > > ### Comment · Reviewer_qbz8 · 2024-11-28
> > >
> > > Thank you for your detailed explanation and the extensive efforts you have put into this work. Given the depth of the analysis, I am raising my score.
> > >
> > > However, I would like to suggest a few clarifications to avoid potential overclaiming. For instance, the statement *"we observe that Frame-Voyager can generalize to 160 frames"* may need some refinement. Is it Frame-Voyager combined with LLaVA-OneVision, or possibly the underlying Qwen2 model, that enables generalization to 160 frames? It would be helpful to clarify whether the original Frame-Voyager model with VILA-8B can also generalize to this number of frames.
> > >
> > > Additionally, the claim *"the results confirm Frame-Voyager's effectiveness and flexibility in dynamically selecting frames"* could benefit from further elaboration. If Frame-Voyager indeed excels in dynamically selecting frames, one might expect its performance to outperform the `Fixed Num.` baseline, since different types of questions likely require varying amounts of visual information.
> > >
> > > Overall, I encourage the authors to carefully consider these points and carefully check their draft to ensure that the conclusions drawn are fully supported by the experimental results.

---

> > > > ### Author Response · Authors · 2024-11-29
> > > > **Appreciation for Your Feedback and Acknowledgment & Third Round Response**
> > > >
> > > > Dear Reviewer qbz8,
> > > >
> > > > Thank you for acknowledging our work and providing thoughtful feedback. We truly appreciate your insights and constructive suggestions, which have helped us improve the paper. All updated results and discussions have been incorporated into the revised version, with changes highlighted in blue.
> > > >
> > > > Below, we provide detailed responses to your follow-up comments:
> > > >
> > > > ***
> > > >
> > > > >**Q1: Effect of first bottom layer in Qwen2 model**
> > > >
> > > > **Response:**
> > > >
> > > > Thanks for raising this important point.
> > > >
> > > > First, we would like to highlight that Frame-Voyager serves as a plug-and-play frame combination selection module that enhances existing Video-LLMs. While it may inherently benefit from superior Video-LLM architectures, the performance suggests its effectiveness is not dependent on any specific Video-LLM backbone.
> > > >
> > > > Second, to evaluate the contribution of Qwen2's first bottom layer, we test Frame-Voyager (trained on VILA-8B) in selecting 160 frames from a 512-frame candidate pool. We maintain the same experimental setting as in ```Response to the Reviewer qbz8 (Second Round-2)```. By directly feeding selected 160 frames into  LLaVA-One-Vision for prediction, we obtain a result of 60.3. It is slightly lower than our previous result of 61.0 (Frame-Voyager also trained on LLaVA-One-Vision), but still outperforms the uniform sampling (58.2).
> > > >
> > > > In conclusion, these results demonstrate that Frame-Voyager's performance improvements stem primarily from its architectural design rather than from Qwen2's frozen features. The module maintains robust performance across various Video-LLMs, regardless of their individual capabilities.
> > > >
> > > > ***
> > > >
> > > >
> > > > >**Q2: Dynamic Num. Compared to Fixed Num.**
> > > >
> > > > We clarify that Frame-Voyager with Dynamic Number demonstrates superior performance compared to Frame-Voyager with Fixed Number when both utilize a similar average number of frames.
> > > > In the ```Response to Reviewer qbz8 (Second Round-2)```, experimental results with different models reveal this advantage:
> > > >
> > > > **With VILA-8B:**
> > > > * Dynamic Number (average 4.1 frames): 49.6
> > > > * Fixed 4 frames: 47.9
> > > >
> > > > **With LLaVA-One-Vision-7B:**
> > > > * Dynamic Number (average 13.3 frames): 59.1
> > > > * Fixed 16 frames: 59.2
> > > >
> > > > The results demonstrate Frame-Voyager's adaptive frame selection capability, maintaining performance comparability while potentially reducing computational overhead.
> > > >
> > > > ***
> > > >
> > > >
> > > > Best regards, Author(s) of submission #13344

---

> > > > > ### Comment · Reviewer_qbz8 · 2024-11-29
> > > > >
> > > > > Is 49.6 the best performance achieved by `Dynamic` Frame-Voyager with VILA-8B? If so, it still falls behind its `Fixed` counterpart, which achieves 50.5 with 8 frames. The same applies to LLaVA-OneVision-7B.
> > > > >
> > > > > Furthermore, how can the claim that “Frame-Voyager with a `Dynamic` Number demonstrates **superior** performance” be justified when it is essentially on par with `Fixed` with LLaVA-OneVision-7B?

---

> > > > > > ### Author Response · Authors · 2024-11-30
> > > > > >
> > > > > > Thank you for the valuable suggestions. To clarify, we perform a preliminary verification during the rebuttal phase to demonstrate its feasibility and potential. In future work, we aim to optimize the dynamic version of Frame-Voyager to further improve its performance.
> > > > > >
> > > > > > We acknowledge that the claim “Frame-Voyager with a dynamic number demonstrates superior performance” may have been overstated. Our intention is to emphasize that the dynamic Frame-Voyager achieves comparable results to the fixed version while using fewer frames (13.3 vs. 16). We appreciate the reviewer’s feedback and will refine the wording in our paper to ensure all conclusions are well-supported by the experimental results.

---

### Official Review · Reviewer_6u8S · 2024-11-01

**Soundness:** 2
**Presentation:** 3
**Contribution:** 2
**Rating:** 6
**Confidence:** 5

**Summary:**

In this paper, the authors present Frame-Voyager which learns to query informative frame combinations, based on the given textual queries for video qa tasks. The authors create a data collection pipeline for the proposed model/training objection and evaluate it on four video QA benchmarks.

**Strengths:**

1. The motivation is clear and easy-to-follow.

2. The presentation of the paper is of high quality.

3. The analysis is comprehensive.

**Weaknesses:**

1. The inflexibility of the proposed frame selection method. Different videos usually have different “information density”, which means that some video could be represented by even a single frame and some need densely sampled frames. This also would be affected by the query type. The number of keyframes in the proposed framework seems to be a fixed hyper-parameter, which would be hard to generalize to all different videos in the wild (for different length / query types, need different hyper-parameter selection). Is it possible for the model to do adaptive/dynamic frame selection?

2. The authors mention in line 170-172 that “the smaller combinations exhibit generalization capabilities when larger values of M and T are used during inference for longer video”. However, the process of this generalization is unclear (the introduction of the inference process, top-K selection is also unclear, in line 258-261). Does the inference directly apply the model to higher frame number or is the concatenation of several combinations with the highest score, what is the K setting for inference? Also, based on this, could the authors elaborate more on how Frame-Voyager makes good use of global video context?

3. In Table 1 and Figure 3, the authors only compare Frame-Voyager with the uniform sampling baseline using the same frame number of frames for Question-Answering. However, Frame-Voyager actually sees more frames for keyframe selection (128 frames and select 8 frames), which creates unfair comparisons. Also there seems to be very little improvement in performance while using 8 keyframes from 128 frames compared to just uniform sample 16 frames. Could the authors show the comparison of Frame-Voyager against the best uniform sampling configurations for a more fair comparison?

**Questions:**

All question in weakness and:

1. Since the authors claim that the proposed method is a plug-and-play module, does the reference Video-LLM for data collection have to be the same model family with the MLLM for Frame-Voyager? Since different models have different features, whether the generated data only works for the ViLA-1.5 model family? It would be interesting to see the performance of these annotated data’s effectiveness on different model families like LLaVA-onevision and so on.

2. The authors are encouraged to discuss the comparison with a related line of research which leverages LLM agents to select meaningful keyframes from the video input, including VideoAgent [1], VideoTree[2] …

3. Also, could the authors validate the effectiveness of the frame selection method on temporal grounding as well? Which seems to be very similar in high-level (localizing key information given the text query).

[1] VideoAgent: Long-form Video Understanding with Large Language Model as Agent

[2] VideoTree: Adaptive Tree-based Video Representation for LLM Reasoning on Long Videos

---

> ### Author Response · Authors · 2024-11-21
> **Response to the Reviewer 6u8S (First Round-1)**
>
> > **W1: Adaptive/dynamic frame selection**
>
> **Response:**
>
> Thank the Reviewer ```6u8S``` for the valuable and insightful suggestion. We agree with the observation about varying information density across different videos and query types. Our Frame-Voyager method can indeed conduct dynamic frame selection without requiring additional training or supervision. This can be achieved by normalizing the rewards of candidate frames to a 0-1 range and selecting frames that exceed a specified threshold.
>
> To validate this capability, we conduct experiments on Video-MME (without subtitles) using VILA-8B. We set a normalized reward threshold of 0.7 and maintain our original constraints (maximum 8 frames, minimum 1 frame) for a fair comparison. The experimental results are shown as follows.
> | **Method**                        | **Num. of Frames (Avg.)** | **Video-MME (w/o subtitles)** |
> |-----------------------------------|--------------------|---------------|
> | Uniform Sampling              | 4                  | 44.7          |
> | Frame-Voyager + Fixed Num.    | 4                  | 47.9          |
> | Uniform Sampling             | 8                  | 47.5          |
> | Frame-Voyager + Fixed Num.    | 8                  | 50.5          |
> | **Frame-Voyager + Dynamic Num.**  |  **4.1**                  | **49.6**          |
>
> From the results, we can find that Frame-Voyager with dynamic selection can improve overall performance while maintaining a similar average number of used frames. Specifically, the fixed-number selection (4-frame combination) achieves a result of 47.9, whereas the dynamic-number selection achieves a result of 49.6, with an average of 4.1 frames. **This demonstrates that Frame-Voyager is also capable of conducting dynamic frame selection.**
>
> Building on this experiment, we further examine the required information density (defined by the number of dynamically selected frames in our model) from three perspectives: video duration, video content domains, and query types. We present our experimental results below.
>
> | **_Video Duration_** | **Num. of Dynamically Selected Frames (Avg.)** |
> |----------------------|---------------------------|
> | Short           | 3.72                      |
> | Medium           | 4.07                      |
> | Long             | 4.61                      |
>
>
> | **_Video Content Domain_**       | **Num. of Dynamically Selected Frames (Avg.)** |
> |--------------------------|---------------------------|
> | Knowledge            | 3.89                      |
> | Film & Television    | 3.96                      |
> | Sports Competition   | 4.43                      |
> | Artistic Performance  | 3.95                      |
> | Life Record          | 3.97                      |
> | Multilingual         | 4.31                      |
>
>
> | **_Query Type_**         | **Num. of Dynamically Selected Frames (Avg.)** |
> |--------------------------|---------------------------|
> | Perception           | 3.01                      |
> | Recognition          | 3.84                      |
> | Reasoning            | 4.58                      |
> | OCR                  | 3.93                      |
> | Counting             | 5.36                      |
> | Information Synopsis  | 4.35                      |
>
> The results reveal some patterns in how our dynamic selection adapts to different scenarios:
> * Video Duration: Longer videos naturally require more frames (Short: 3.72, Medium: 4.07, Long: 4.61 frames on average).
> * Video Content Domains: The correlation between frame requirements and video domains is less obvious. Sports videos require the most average frames (4.43) due to frequent action changes, while other domains maintain relatively consistent frame requirements (3.89-4.31).
> * Query Types: Task complexity does influence frame selection. Complex tasks like counting (5.36 frames) and reasoning (4.58 frames) require more frames for comprehensive analysis. Note that counting tasks, despite having lower reasoning demands, usually requires answering questions like "How many different people appear in the video?", which strongly depend on multiple frames. Simpler tasks like perception (3.01 frames) and recognition (3.84 frames) need fewer frames.
>
> **The results confirm Frame-Voyager's effectiveness in dynamically selecting frames by considering both the video's information density and the specific query needs**. We will further include these bonus results and analysis in the revised version.

---

> > ### Comment · Reviewer_6u8S · 2024-11-23
> > **Response to the Author response First Round-1**
> >
> > Thanks the authors for the detailed response. Overall I like these results, but I think I still have some follow-up question:
> >
> > 1. In this dynamic frame selection, does the model still considering combination of it's more like separate frames and decide the selection of each frame by a threshold (I understand the generated final query feature Y_FFN contains global information, but since the reward is just a cosine calculation, the fuse of the global information is still limited).
> >
> > 2. The results for the dynamic selection on long split looks weird, since the videos are 30-60 minutes long and it seems the authors claim that less than 5 frames is already enough (also the results look similar to the uniform 16 frame results in Figure 3). Also the current  Could the author use the llava-ov backbone (which availble for more frames) to further verify the effectiveness of the proposed dynamic method on a higher frame rate?

---

> > > ### Author Response · Authors · 2024-11-23
> > > **Response to the Reviewer 6u8S (Second Round-1)**
> > >
> > > Thank you for your follow-up questions and comments. We truly appreciate the time and effort you’ve taken to engage with our work in such detail. Your feedback has been very valuable in helping us improve the manuscript.
> > >
> > > Below, we provide detailed responses to your follow-up queries:
> > >
> > > ***
> > >
> > > > **Q1: Combination or separate frames in dynamic frame selection**
> > >
> > > **Response:**
> > >
> > > Frame-Voyager still incorporates frame combination modeling in the dynamic frame selection. We believe that the Frame-Voyager module has been well trained via LLM parameters and combination reward optimization to capture the global context. Furthermore, we emphasize that threshold-based adaptive sampling presents a straightforward yet effective approach for inference using the current model. The experimental results (detailed in ```Response to Reviewer 6u8S (First Round-1)```) substantiate both the feasibility and effectiveness of this approach.
> > >
> > > The dynamic characteristics of Frame-Voyager can be further enhanced across various algorithm designs. For instance, future work could explore varying the frame count during data collection and developing modules capable of predicting the optimal number of frames required for a given (video, query) pair.
> > >
> > > ***
> > >
> > > > **Q2: Explanation and more results on dynamic selection**
> > >
> > > **Response:**
> > >
> > > First, we would like to clarify several points regarding the dynamic frame selection analysis (presented in ```Response to Reviewer 6u8S (First Round-1)```) .
> > > * We set the threshold to 0.7 while applying the constraints to avoid too many or too few frames, i.e., maximum 8 frames, minimum 1 frame. Our analysis is conducted under the condition above, thus the number of frames is limited.
> > > * Moreover, the variance of the number of frames allocated for long videos is 3.6 and the maximum number of frames is 8. It suggests that Frame-Voyager allocates more than 5 frames (or even more than 8 frames) to some of the long videos. We also provide the percentage of samples assigned with 8 frames by Frame-Voyager, which accounts for 19.4% in short videos and 31.3% in long videos.
> > >
> > >
> > > Second, in response to Reviewer ```6u8S```, we conduct additional experiments on LLaVA-One-Vision. For these experiments, we expand the maximum frame limit to 32 and adjust the threshold to 0.5 to accommodate a larger frame selection. The results demonstrate the effectiveness of Frame-Voyager with a higher number of frames, i.e., Frame-Voyager achieves a score of 59.1 using 13 frames, surpassing the performance of uniform sampling with 32 frames.
> > >
> > > | LLaVA-One-Vision                 | **Num. of Frames (Avg.)** | **Video-MME (w/o subtitles)** |
> > > |----------------------------------|---------------------------|-------------------------------|
> > > | Uniform Sampling                 | 8                         | 53.3                          |
> > > | Frame-Voyager + Fixed Num.       | 8                         | 57.5                          |
> > > | Uniform Sampling                 | 16                        | 57.2                          |
> > > | Frame-Voyager + Fixed Num.       | 16                        | 59.2                          |
> > > | Uniform Sampling                 | 32                        | 58.2                          |
> > > | Frame-Voyager + Fixed Num.       | 32                        | 60.4                          |
> > > | **Frame-Voyager + Dynamic Num.** | **13.3**                  | **59.1**                      |
> > >
> > >
> > > ***
> > >
> > >
> > > \
> > > \
> > > \
> > > We hope these responses address your concerns. Please let us know if there are any additional questions.

---

> > > > ### Comment · Reviewer_6u8S · 2024-11-27
> > > >
> > > > Thanks the authors for the detailed response. I like the updated results and would like to see the results incorporate to the final draft. I appreciate the effort from the authors and willing to raise the score to 6.

---

> > > > > ### Author Response · Authors · 2024-11-28
> > > > > **Appreciation for Your Feedback and Acknowledgment!**
> > > > >
> > > > > Dear Reviewer 6u8S,
> > > > >
> > > > > Thank you for acknowledging our work and providing thoughtful feedback. We truly appreciate your insights and constructive suggestions, which have helped us improve the paper. All updated results and discussions have been incorporated into the revised version, with changes highlighted in blue.
> > > > >
> > > > > Best regards,
> > > > > Author(s) of submission #13344

---

> ### Author Response · Authors · 2024-11-21
> **Response to the Reviewer 6u8S (First Round-2)**
>
> > **W2.1: Generalization process is unclear**
>
> **Response:**
>
> It is “directly apply the model to a higher frame number”. The definition of the generalization here is: when training, we sample 4 frames from 32 frames (or 2 frames from 16 frames), and when inference, we directly choose 8 frames from 128 frames. These numbers are different, so it is generalized from smaller numbers to larger numbers.
>
> Additionally, for more analysis and results regarding this generalization process, please kindly refer to the ```General Response of First Round Rebuttal (2/4)``` comment. In this general response, we explain why Frame-Voyager is able to generalize to long videos and larger numbers of frames. We also add experimental support based on the LLaVA-One-Vision.
>
> As for the parameter K, we believe the Reviewer ```6u8S``` has confused between K and T. The K is only used in training, and it means the number of selected combinations for ranking loss optimization. The T is used both in training and inference, and it means the frame number in each combination.
>
> ***
>
> > **W2.2: How Frame-Voyager makes good use of global video context**
>
> **Response:**
>
> Frame-Voyager is built upon the selection of frame combinations instead of retrieving individual frames separately, and it naturally requires a good understanding of the global context of videos (Lines 54-56). In terms of the question of “How Frame-Voyager makes good use of global video context”, we believe there are two points:
>
> (1) For the model architecture of  Frame-Voyager, we feed all the candidate frames and utilize the bottom layers of the LLM whose attention mechanism enables the simultaneous processing of frame-to-frame and frame-to-query relationships.
>
> (2) For the training of Frame-Voyager, we design a ranking loss (Equation 3, Lines 250-253) to optimize the selection of frame combinations. It further encourages the model to capture global information for identifying the better combination regarding the given query.
>
> ***
>
> > **W3: Comparison fairness**
>
> **Response:**
>
> Thank the Reviewer ```6u8S``` for raising this point about comparison fairness. We would like to clarify two key aspects of our experimental design and present additional results that demonstrate our method's effectiveness.
>
> (1) **First, regarding fairness of comparison**: while our Frame-Voyager initially observes 128 frames, **it ultimately uses only 8 frames for final processing by the LLM, similar to all the baselines**. Both methods have access to the full video - the difference lies in how they select frames. Uniform sampling uses a straightforward distribution strategy, while our method employs selection based on the modeling of the global context between the video and the query. We also compare our method against several frame selection baselines (rather than uniform sampling), as shown in Table 2 (Lines 324-337). These baselines include:
>
>   - Rule-based shot boundary detection methods: Histogram, Edges Change Ratio, and Motion, which select frames based on significant frame transitions in texture, structure, and motion.
>   - Cluster-based method: This approach extracts histograms from all frames and employs K-means clustering to select the most representative frames near cluster centers.
>   - Frame-text matching methods: These two methods retrieve frames by calculating cosine similarity between query inputs and individual frames using VILA and CLIP embeddings.
>
> The results demonstrate that the **Frame-Voyager outperforms all these baseline methods**, showing superior performance in both frame subset selection and efficient video understanding.
>
> (2) **Second, regarding best uniform sampling configuration**: due to the limit of VILA-8B, please kindly refer to ```General Response of First Round Rebuttal (2/4)``` for more results conducted on LLaVA-One-Vision. We could observe that **Frame-Voyager shows consistently better performance when increasing the number of frames from 2 to 64 (with a best performance of 60.5%). The uniform sampling method plateaus at 32 frames (with a best performance of 58.2%)**.

---

> ### Author Response · Authors · 2024-11-21
> **Response to the Reviewer 6u8S (First Round-3)**
>
> > **Q1: Results on LLaVA-One-Vision**
>
> **Response:**
>
> Please kindly refer to ```​​General Response of First Round Rebuttal (1/4)``` for the results of Frame-Voyager on LLaVA-One-Vision trained with the same data generated by VILA-8B.
>
> From the results, it is shown that Frame-Voyager trained with data generated by one Video-LLM (in our case, VILA-8B) can be zero-transferred to another Video-LLM, e.g., LLaVA-One-Vision-7B. **This demonstrates the generalizability and plug-and-play ability of Frame-Voyager.**
>
> ***
>
> > **Q2: Comparison with LLM agents**
>
>
> **Response:**
>
> We appreciate the reviewer highlighting these related papers. We would like to highlight that the two agent-based methods are not directly comparable to Frame-Voyager due to the large language models and complex interaction steps they have, **such as multi-round interactions with an LLM-agent and the use of closed-source GPT-4 as the final reasoning tool**. We elaborate more on the key conceptual differences as follows.
>
> 1. VideoAgent's approach begins with uniformly sampled frames and uses CLIP for text-based frame retrieval. This differs from Frame-Voyager's comprehensive frame combination strategy, which considers the relationships between multiple frames rather than relying solely on text matching of VideoAgent.
>
> 2. VideoTree employs a hierarchical approach, first clustering frames and then using language model analysis of cluster captions to guide further exploration by GPT-4. While this method implicitly considers frame relationships through caption analysis, Frame-Voyager takes a more direct approach by explicitly evaluating and ranking complete frame combinations.
>
> Note that both methods rely on running complete Video-LLM models multiple times, requiring significant computational resources. **These comparisons highlight Frame-Voyager's unique contribution: a streamlined, efficient approach to frame selection that directly optimizes for informative frame combinations rather than relying on intermediate steps or external reasoning tools.** We will also include the appropriate citations and discussions in our revised manuscript.
>
> References:
>
> [1] VideoAgent: Long-form Video Understanding with Large Language Model as Agent
>
> [2] VideoTree: Adaptive Tree-based Video Representation for LLM Reasoning on Long Videos
>
> ***
>
> > **Q3: Experiment on temporal grounding**
>
>
> **Response:**
>
> Thank the Reviewer ```6u8S``` for this suggestion. While we appreciate the recommendation to evaluate our method on grounded QA (temporal grounding), it's important to note that Grounded QA and our approach address different tasks of video understanding.
> * Grounded QA: focuses on identifying specific continuous temporal segments directly related to a question. It’s more suitable for local questions.
> * Our approach:  focuses on selecting multiple frames with temporal relations (but no need to be continuous) that capture broader temporal relationships across the video. It allows us to handle both local and global questions.
> To provide a fair comparison, as mentioned in the response to W3, we adapt the grounding method in TempGQA [1] to identify relevant video segments based on the question, and then sample frames from these segments for answer generation with VILA-8B. On the VideoMME (without subtitles) benchmarks, the results are:
>
>   - TempGQA-based approach: 46.4;
>
>   - Baseline uniform sampling with VILA-8B: 47.5;
>
>   - Our Frame-Voyager method: 50.5.
>
> These results demonstrate that while grounding approaches are valuable for their specific use case, **our method's broader frame combination strategy achieves better performance on general video understanding tasks.** We will add this additional baseline method in the revised manuscript.
>
> [1] Can i trust your answer? visually grounded video question answering. CVPR 2024.
> ***
>
> \
> \
> \
> Please kindly let us know if you have any further questions or comments. We would be happy to address them.

---

### Official Review · Reviewer_pMFu · 2024-11-04

**Soundness:** 3
**Presentation:** 2
**Contribution:** 3
**Rating:** 6
**Confidence:** 4

**Summary:**

This paper proposed Frame-Voyager that learns to query informative frame combinations, based on the given textual queries in the task.
Authors introduced a new data collection and labeling pipeline, by ranking frame combinations using a pre-trained Video-LLM.
Extensive experiments are conducted to support the effectiveness of the proposed Frame-Voyager.

**Strengths:**

This is a reasonable extension from previous keyframe selection work [1] where it relies more on single keyframe selection, and does not consider temporal relations/modeling among frames. The proposed pseudo-label scheme from a VLM + list of combinations of frames makes sense to me.
Authors also conduct extensive experiments across diverse popular benchmarks to show their effectiveness.
Overall, I think the proposed Frame-Voyager makes a good contribution to the keyframe selection in video-language studies.

[1] Self-chained image-language model for video localization and question answering. NeurIPS23

**Weaknesses:**

1. As the paper mentioned, this pseudo-label strategy is not scalable, and I think the frame combination part is a bit tricky. This is like creating artificial rewards according to video (like a sandbox in RL) to train the reward model. However,  the reward is not always reliable from a VLM even though we compute it according to GT answers. For example, as shown in some previous studies [2],  the model will generate correct answers when provided with wrong localized clips. It is true that we might give performance improvement when using such reward data created by a human-free pipeline, but it is more like an adapter / DPO style adaptation to the model rather than letting the model truly be grounded on query-related / informative frames.

2. I think the weakness in 1 somehow affects the proposed method to require a relatively complex pre-training data construction/filtering/design, since the positive/negative signal is too sensitive according to video conditions in this framework from my view (e.g. frame blurring / redundancy).

3. Also, the keyframe selection for video-language understanding is not a brand new topic, many related works in track try to propose different ways, from continuous space learning to discrete pipeline, to address question-aware moment detection. I would suggest to include those works [1,2,3,4] for a more comprehensive study.


[1] ViLA: Efficient Video-Language Alignment for Video Question Answering. ECCV24.
[2] Can i trust your answer? visually grounded video question answering.  CVPR24.
[3] TimeCraft: Navigate Weakly-Supervised Temporal Grounded Video Question Answering via Bi-directional Reasoning. ECCV24.
[4] Self-Adaptive Sampling for Accurate Video Question Answering on Image Text Models. ACL24.

**Questions:**

Please see the weaknesses. And extra questions:

(1) What is the trade-off between efficiency and effectiveness? Can you provide some metrics like running time/memory usage/flops to the proposed methods? As the model contains an extra keyframe localization stage, it is worthy showing those results to see the trade-off.

(2) I notice that the proposed method uses VILA as a reference model to label rewards, and uses the same VILA series models for downstream tasks. Is this also a kind of self-rewarding strategy shown in the sevilla work? Or the proposed method/modules can zero transfer to other VLM like llava-ov/qwen-vl-2?

(3) to answer my 1st weakness, I would suggest authors conduct extra experiments on Next-GQA to see the grounded QA results with grounded metrics.

(4) Can you show some off-shelf llm-based localization tools like sevila localizer in Table 2? It would be interesting to see the comparison between a llm-based reasoning method with the proposed reward imitation learning method.

(5)  Regarding the claim in Figure 3 and RQ3 (Lines 431-433), is this true? Related work (e.g., [1]) suggests that some observed issues may stem from limitations in the model side (VILA). Could the authors clarify this?

[1] LONGVIDEOBENCH: A Benchmark for Long-context Interleaved Video-Language Understanding

---

> ### Author Response · Authors · 2024-11-21
> **Response to the Reviewer pMFu (First Round-1)**
>
> > **W1: Reliability of the pseudo-label strategy**
>
> **Response:**
>
> Thank the Reviewer ```pMFu``` for raising these important concerns. We agree that in certain cases LLM can intuitively generate correct answers even with wrong localized frames, primarily due to inherent biases in language models such as language shortcuts and spurious vision-language correlations [1].
>
> We already consider this biased issue of video-LLM, as mentioned in Lines 173-177. To avoid this, our method incorporates the following designs, each compared with the alternative method strongly affected by such bias:
>
> 1. **We implement necessary filterings against language model biases in our data preparation pipeline** (Lines 175-177). We analyze the variance in losses and find that pairs with low variance in the losses across different combinations are not sensitive to the quality of combinations. One of the reasons might be that correct answers may be generated based on the Video-LLM’s inherent language prior without referring to any input content. Thus, we exclude these cases where the model appears to rely primarily on language priors. The effectiveness of this filtering step is validated through ablation studies, as shown in Table 3 (Lines 331-332), comparing VideoChatGPT with and without filtering (49.1 vs 48.3).
>
> 2. **Our framework adopts a comparative approach that ranks different frame combinations based on their relative effectiveness**, rather than aiming to identify a single "perfect" combination, which is usually infeasible.
> *  Instead of using binary rewards (correct/incorrect answers) or selecting only the lowest-loss combination, we implement a ranking-based loss function that compares pairs of frame combinations. From the perspective of RL, this continuous reward structure provides more stable training signals compared to discrete alternatives [4].
> * Our data collection method goes beyond simple frame-question pairs for answer prediction, which cannot differentiate between the inherent biases of LLMs and the usage of provided frame combinations. Instead of directly evaluating answer correctness, we use the internal loss to measure the model's confidence during the answer generation. This loss reflects the model's confidence [2, 3] in producing the answer given the frame combination and question, where lower loss indicates higher confidence.
> * Our experimental results in Table 3 (Lines 333-336) demonstrate that this comparative approach (Lines 334-336, setting (5-6) and Frame-Voyager with K=4) consistently outperforms methods trained to select only the single best frame combination (Line 333, setting (4) Top-1 Rank).
>
> **In summary, our method produces reliable training signals while minimizing the impact of language model biases.** We include additional ranking data examples in this [anonymous link](https://anonymous.4open.science/r/iclr_rebuttal-2450/ranking_case.jpg) to further demonstrate the quality of our generated training data for reference.
>
> [1] Can i trust your answer? visually grounded video question answering. CVPR 2024.
>
> [2] On Calibration of Modern Neural Networks. ICML 2017.
>
> [3] Language Models (Mostly) Know What They Know. ArXiv 2022.
>
> [4] Reward function design in reinforcement learning. Eschmann, J. Reinforcement Learning Algorithms: Analysis and Applications 2021.
>
> ***
>
> > **W2: Complex pre-training data design due to the sensitive signal**
>
> **Response:**
>
> Based on the specialized design mentioned in the response of W1, we believe that the quality of the generated ranking data is guaranteed. Meanwhile, we also observe that the combinations with redundancy or blurring frames tend to have lower ranking due to the lack of information. For example, we can observe that the combination with similar frames has lower rank in the second case shown in the [anonymous link](https://anonymous.4open.science/r/iclr_rebuttal-2450/ranking_case.jpg).

---

> > ### Author Response · Authors · 2024-11-21
> > **Response to the Reviewer pMFu (First Round-2)**
> >
> > > **W3: Include more work for a more comprehensive study.**
> >
> > **Response:**
> >
> > Thank the Reviewer ```pMFu``` for highlighting these relevant papers. We will incorporate them into our literature review and provide more comparisons with our work. Our method differs from these methods in several key aspects as follows:
> > * Comparison with ViLA [1] and Self-Adaptive Sampling [4]: our approach differs fundamentally in how we conduct the frame selection. While we explicitly model frame combinations, [1] focuses on individual frame-question relationships without considering inter-frame dynamics. Similarly, [4] selects frames based only on visual features, without considering textual question information. Given the time limitations of the rebuttal stage, we choose to reproduce the method in [4] for our ablation studies and plan to include a comparison with [1] in the camera-ready version. Specifically, we implement [4]'s method using their open-source code and default parameters with VILA-8B's visual encoder for a fair comparison. **Using the VILA-8B backbone, the method [4] achieves 47.8 on Video-MME (without subtitles), compared to our score of 50.5. Additionally, the method [4] uses an average of ~11 frames (dynamically selected), our approach achieves better results with only 8 frames.**
> > * Comparison with Grounded VQA approaches [2, 3]: we think that these methods address different tasks of video understanding.
> >   - The grounded VQA focuses on identifying specific continuous temporal segments directly related to a question. It’s more suitable for local questions, and lacks the ability to solve tasks requiring global information. In contrast, our method selects optimal frame combinations that can address both local and global video understanding tasks.
> >   - To make a comparison, we adapt the Grounded VQA approach to our video understanding benchmarks used in the paper. We reproduce the grounding method TempGQA in [2] to select a specific grounding segment based on the question. Then we uniformly sample frames from the selected segment and feed them into VILA-8B with questions to generate answers. **The results show that the TempGQA-based method performs a score of 46.4 on VideoMME (without subtitles) while the original uniform sampling of VILA-8B achieves 47.5. This may be because the grounding-based method has difficulty handling questions requiring global information, such as summarization. Our method achieves a higher score of 50.5 under this setting.**
> >
> >
> > [1] ViLA: Efficient Video-Language Alignment for Video Question Answering. ECCV 2024.
> >
> > [2] Can i trust your answer? visually grounded video question answering. CVPR 2024.
> >
> > [3] TimeCraft: Navigate Weakly-Supervised Temporal Grounded Video Question Answering via Bi-directional Reasoning. ECCV 2024.
> >
> > [4] Self-Adaptive Sampling for Accurate Video Question Answering on Image Text Models. ACL 2024.
> >
> > ***
> >
> > > **Q1: Efficiency of the Method**
> >
> > **Response:**
> >
> > Please kindly refer to the ```General Response of First Round Rebuttal (4/4)``` for the analysis of overall efficiency.
> >
> > In summary, the Frame-Voyager only introduces a tiny computational overhead. The minimal increases in parameter count and memory usage suggest that our method scales efficiently while achieving superior video understanding capabilities.

---

> > > ### Author Response · Authors · 2024-11-21
> > > **Response to the Reviewer pMFu (First Round-3)**
> > >
> > > > **Q2.1: Is Frame-Voyager a self-rewarding method?**
> > >
> > > **Response:**
> > >
> > > Thanks for raising this important distinction. Our Frame-Voyager fundamentally differs from self-reward strategies like those used in SeViLA [1]. Specifically, the Frame-Voyager demonstrates two key characteristics:
> > >
> > > * Independent Training: Unlike Sevilla's localizer, which requires joint and iterative training with its answering module, our frame selector is trained separately using reference signals from existing Video-LLMs.
> > > * Model Flexibility: The Video-LLM used for generating training signals can be different from the one where Frame-Voyager is ultimately deployed. For instance, our experiments show successful transfer when using VILA-8B as the reference model for training Frame-Voyager, which is then deployed with VILA-40B (Lines 312-313). We further show that this flexibility can be zero-transferred to other Video-LLM families such as LLaVA-One-Vision (the following response to Q2.2).
> > >
> > > In conclusion, these design choices ensure greater flexibility and transferability than self-reward methods. Our Frame-Voyager can potentially be adapted to work with various video understanding models beyond just the VILA family.
> > >
> > > [1] Self-Chained Image-Language Model for Video Localization and Question Answering, NeurIPS 2023.
> > >
> > > ***
> > >
> > > > **Q2.2: Method Zero transfer to other VLM**
> > >
> > > **Response:**
> > >
> > > Please kindly refer to ```​​General Response of First Round Rebuttal (1/4)``` for the results of Frame-Voyager on LLaVA-One-Vision trained with the same data generated by VILA-8B.
> > >
> > > From the results, it is shown that Frame-Voyager trained with data generated by one Video-LLM (in our case, VILA-8B) can be zero-transferred to another Video-LLM, e.g., LLaVA-One-Vision-7B. This demonstrates the generalizability and plug-and-play ability of Frame-Voyager.
> > >
> > > ***
> > >
> > > > **Q3: Experiment on grounded QA tasks**
> > >
> > > **Response:**
> > > Thank the Reviewer ```pMFu``` for this suggestion. While we appreciate the recommendation to evaluate our method on grounded QA, it's important to note that Grounded QA and our approach address different tasks of video understanding.
> > > As we mentioned in the response to W3,
> > > * Grounded QA: focuses on identifying specific continuous temporal segments directly related to a question. It’s more suitable for local questions.
> > > * Our approach:  focuses on selecting multiple frames with temporal relations (but no need to be continuous) that capture broader temporal relationships across the video. It allows us to handle both local and global questions.
> > > To provide a fair comparison, as mentioned in the response to W3, we adapt the grounding method in TempGQA [1] to identify relevant video segments based on the question, and then sample frames from these segments for answer generation with VILA-8B. On the VideoMME (without subtitles) benchmarks, the results are:
> > >
> > >   - TempGQA-based approach: 46.4;
> > >
> > >   - Baseline uniform sampling with VILA-8B: 47.5;
> > >
> > >   - Our Frame-Voyager method: 50.5.
> > >
> > > These results demonstrate that while grounding approaches are valuable for their specific use case, **our method's broader frame combination strategy achieves better performance on general video understanding tasks.** We will add this additional baseline method in the revised manuscript.
> > >
> > > [1] Can i trust your answer? visually grounded video question answering. CVPR 2024.
> > >
> > > ***
> > >
> > > > **Q4: Compare with SeViLA localizer**
> > >
> > > **Response:**
> > >
> > > Thank the Reviewer ```pMFu``` for this suggestion. We conduct additional experiments comparing our approach with SeViLA. SeViLA operates by computing frame scores through FlanT5, evaluating each frame individually against the question prompt. **Using SeViLA’s frame selection (the localizer in SeViLA) and maintaining original hyperparameter settings, we achieve a score of 49.3, which falls short of our Frame-Voyager's performance (50.5).**
> > >
> > > There are two main differences between SeViLA and Frame-Voyager.
> > >
> > > 1) Regarding the model. SeViLA evaluates individual frames independently, and our Frame-Voyager directly evaluates frame combinations.
> > >
> > > 2) Regarding the inference efficiency. SeViLA requires running a full-parameter LLM (i.e., FlanT5 with 3B parameters) during the frame selection stage, introducing notable computational overhead. In contrast, our Frame-Voyager is lightweight (only 20M extra parameters) for frame combination selection.
> > >
> > > We will include this comparison with LLM-based localization tools in the revised version.

---

> ### Author Response · Authors · 2024-11-21
> **Response to the Reviewer pMFu (First Round-4)**
>
> > **Q5: Observed issues may stem from limitations in the model side (VILA)**
>
> **Response:**
>
> Thank the Reviewer ```pMFu``` for this valuable discussion and feedback. Please kindly refer to ```General Response of First Round Rebuttal (2/4)``` for the comprehensive analysis regarding the generalization of Frame-Voyager on higher numbers of frames.
>
> We agree with the Reviewer ```pMFu``` that model capability is another factor contributing to the pattern in Figure 3, and our current explanations (Lines 413-433) is not completed. We will revise this experimental analysis part in the manuscript.
> ***
>
>
> \
> \
> \
> Please feel free to let us know if you have any additional follow-up questions, and we would be happy to address them further.

---

> ### Author Response · Authors · 2024-11-23
> **Kind Reminder: Seeking Reviewer Feedback for Author/Reviewer Discussion Phase**
>
> Dear Reviewer pMFu,
>
> We hope this message finds you well. We would like to express my gratitude for the time and effort that you have dedicated to the review process of our submission.
>
> Furthermore, we are writing to kindly remind you that the rebuttal period is coming to a close. Your feedback and evaluation are crucial to the progress of our work, and we value your expertise and insights immensely.
>
> During rebuttal, we have thoroughly addressed all the queries and concerns you raised regarding our submission. Each point has been carefully considered, and we have provided detailed responses.
>
> We are eager to engage in further discussions with you on the responses and any other aspects of our study. Your feedback plays a pivotal role in shaping the direction of our research, and we are keen to hear your thoughts and suggestions.
>
> Thank you for your attention to this matter.
>
> Best regards, Authors of submission #13344

---

> > ### Author Response · Authors · 2024-11-25
> > **Kind Reminder: Seeking Reviewer Feedback for Author/Reviewer Discussion Phase**
> >
> > Dear Reviewer pMFu,
> >
> > We regret any disturbance this message may cause and appreciate your understanding.
> >
> > We have posted our responses to address your concerns.
> >
> > With **less than two days remaining** in the discussion period, we kindly hope we can get your feedback.
> >
> > We understand that this is quite a busy period, but we’d appreciate it very much if you could take some time to read our responses. If there are other comments or concerns, we will be happy to address them during the response period.
> >
> > Regards,
> >
> > Author(s) of submission #13344

---

> > > ### Comment · Reviewer_pMFu · 2024-11-28
> > > **Thanks for your rebuttal**
> > >
> > > I appreciate the authors for providing a detailed rebuttal. After reviewing the response, I am happy to note that most of my original concerns (Q1, Q2, Q4, and Q5) have been mostly addressed.
> > >
> > > However, I still have some remaining questions and concerns regarding W3/Q3,
> > >
> > > Remaining questions/concerns:
> > > In your response in W3/Q3
> > > > The grounded VQA focuses on identifying specific continuous temporal segments directly related to a question. It’s more suitable for local questions, and lacks the ability to solve tasks requiring global information.
> > >
> > > > In contrast, our method selects optimal frame combinations that can address both local and global video understanding tasks.
> > >
> > > > it's important to note that Grounded QA and our approach address different tasks of video understanding. As we mentioned in the response to W3,
> > >
> > > I found this explanation intriguing but slightly unclear. From my understanding, learning to answer a question grounded in a specific temporal span in the video still requires global video understanding. This is because effectively grounding an answer often involves modeling relationships across different temporal spans, even if the final response is localized to a specific segment.
> > >
> > > Furthermore, in datasets like Video-MME, especially on the long-video split, many questions do not necessitate a holistic or complete understanding of the entire video. For example, even in videos that are 30 minutes to an hour long, human annotators often answer questions by focusing on very short (1-3 minute) visual segments or even single moments. This suggests that holistic/global frame sampling might not always be a strict prerequisite for answering long VQA effectively, but grounding can help more.
> > >
> > > From my perspective, learning to query frames would be particularly valuable if it faithfully aligns with temporal grounding from a human perspective. Without this alignment, there’s a risk that the querying process becomes an optimization step to adapt to model-specific constraints (e.g., deblurring inputs or fine-tuning frame selection heuristics).
> > >
> > > To demonstrate the value of your approach, I believe it would be important to evaluate your results against grounded metrics or benchmarks that assess human-aligned temporal grounding (which I have suggested in the initial review). Datasets like Next-GQA, which provide human-annotated query-related spans, could offer valuable insights into what is being learned during the "learning to query frames" process.
> > >
> > > I appreciate the novelty of the method, I feel that a clearer justification for its alignment with human-grounded video understanding tasks would strengthen the work. Reporting comparisons against grounding-based metrics would also address concerns about whether the querying mechanism is genuinely learning human-aligned temporal reasoning or simply optimizing for task-specific performance.

---

> > > > ### Author Response · Authors · 2024-11-30
> > > > **Response to the Reviewer pMFu (Second Round)**
> > > >
> > > > Thank you for your follow-up questions and comments. We truly appreciate the time and effort you’ve taken to engage with our work in such detail. Your feedback has been very valuable in helping us improve the manuscript.
> > > >
> > > > ***
> > > >
> > > > We fully agree with the Reviewer ```pMFu``` that evaluating results against grounded metrics would strengthen the work. To adapt keyframe selection methods for grounded video tasks, we arrange extracted keyframes chronologically and define the grounding result as the span from the first to the last frame. Since the uniform sampling method covers the entire video in this way, which is not suitable for this task. Instead, we compare Frame-Voyager with CLIP and SigLIP methods.
> > > >
> > > > Following the NextGQA paper [1], we report mIoP and mIoU metrics. Our results demonstrate that Frame-Voyager significantly outperforms CLIP and SigLIP across both grounding metrics. Empirically, we observe that CLIP and SigLIP struggle in localizing groundings of temporal reasoning tasks, which accounts for about 20% in NextGQA.
> > > >
> > > > | NextGQA       | **mIoP** | **mIoU** |
> > > > |---------------|----------|----------|
> > > > | CLIP          | 24.5     | 15.3     |
> > > > | SigLIP        | 26.1     | 18.9     |
> > > > | Frame-Voyager | 31.3     | 22.6     |
> > > >
> > > > Furthermore, we wish to emphasize that keyframe selection methods may not achieve SOTA performance on grounded metrics as it is not explicitly optimized for these objectives. Nevertheless, compared with baseline methods, the significant improvements on mIoP and mIoU help confirm the Frame-Voyager's effectiveness in selecting relevant frame combinations for video-QA. We will incorporate these additional grounding metrics in our revised manuscript.
> > > >
> > > > [1] Can i trust your answer? visually grounded video question answering. CVPR 2024.

---

> > > > > ### Comment · Reviewer_pMFu · 2024-12-01
> > > > > **Thanks for new results on grounded QA**
> > > > >
> > > > > My concerns on grounding metrics are well-addressed. I appreciate those results. I would highly suggest incorporating those new results in revision to provide more insights for aligning the model with humans for both reasoning and grounding. I have no other questions for the paper, and I will increase my score to 6.

---

> > > > > > ### Author Response · Authors · 2024-12-01
> > > > > > **Appreciation for Your Feedback and Acknowledgment!**
> > > > > >
> > > > > > Dear Reviewer pMFu,
> > > > > >
> > > > > > Thank you for taking the time to review our work and for providing thoughtful and constructive feedback. We sincerely appreciate your insights, which will be instrumental in improving the quality of our paper.
> > > > > >
> > > > > > We will carefully revise the manuscript and include updated empirical results, such as grounding tasks, along with our discussions as outlined above.
> > > > > >
> > > > > > Thank you once again for your valuable contributions.
> > > > > >
> > > > > > Best regards, Author(s) of submission #13344

---

### Author Response · Authors · 2024-11-20
**General Response of First Round Rebuttal (1/4)**

We thank all the reviewers for their valuable feedback! In this post, **we would like to first address the common questions raised by the reviewers with additional experimental results. In the sooner individual replies, we will address other specific comments.**

**1.** ```[pMFu, 6u8S]``` **Generalization on Different Backbones**: Could our method and generated data be generalized to other Video-LLMs such as LLaVA-One-Vision?
To answer this question, we directly apply the same data obtained by VILA-8B to train Frame-Voyager on one of the SOTA Video-LLMs, LLaVA-One-Vision-7B. We evaluate the models on the same benchmarks used in our paper. The results are shown in the table below.

|      | **Video-MME (w/o subtitles, %)** | **MLVU (%)** | **ANQA (%)** | **NextQA (%)** |
|---------------------|---------------|----------|----------|------------|
| LLaVA-One-Vision-7B | 53.3          | 58.5     | 41.7     | 72.5       |
| **+Frame-Voyager**  | 57.5          | 65.6     | 48.4     | 73.9       |

From the table, we can observe that the Frame-Voyager module can consistently improve the performance of LLaVA-One-Vision-7B across different benchmarks. It shows that Frame-Voyager trained with data generated by one Video-LLM (in our case, VILA-8B) can be zero-transferred to another Video-LLM (in our case, LLaVA-One-Vision-7B). This demonstrates the generalizability and plug-and-play ability of Frame-Voyager.

---

### Author Response · Authors · 2024-11-20
**General Response of First Round Rebuttal (2/4)**

**2.** ```[pMFu, 6u8S, qbz8]``` **Generalization on Higher Numbers of Frames**: How does Frame-Voyager generalize to higher numbers of selected frames?

The generalization on the number of frames in our paper is: when training, we sample 4 frames from 32 frames (or 2 frames from 16 frames), and when inference, we directly choose 8 frames from 128 frames, so it is generalized from smaller numbers to larger numbers. It is aligned with most of the existing Video-LLMs, i.e., trained primarily on short videos, yet they can also generalize to long videos during testing, e.g., VILA and LLaVA-One-Vision [1][2]. In the case of Frame-Voyager, for instance, considering temporal reasoning tasks where the question requires locating the beginning of a video (case study shown in Fig. 6), the Frame-Voyager consistently focuses on the video openings regardless of total duration. Similarly, for content identification tasks, e.g., finding ingredients as shown in Fig. 9, it selects relevant frames containing ingredients independent of video length.

As mentioned in our paper (Lines 430-431), “as the number of extracted frames increases, the performance gap between our method and uniform sampling narrows”. There are mainly two factors, i.e., dataset characteristics and model capacity, that contribute to this pattern observed. We conduct further analysis below for clarification.

1. **Dataset Characteristics**: Our case studies (Fig. 6, Lines 445-485; Fig. 9, Fig. 10, Lines 922-1022) show that 8 or fewer frames selected by Frame-Voyager are sufficient to answer some of the questions. Additional frames may introduce redundant or noisy information for some question types, reducing the model performance (Lines 037–040). This finding is consistent with LongVA's results [3].
2. **Model Capability** [suggested by the Reviewer ```pMFu```]: Given the selected frame combination, the Video Question Answer performance is determined by the inference model, e.g., the frozen VILA used in our paper. For example, in Figure 5 (Lines 402-420), it is shown that VILA is generally not good at addressing the OCR and counting problems using any frame selection methods (uniform sampling, CLIP, or ours).

Due to the model capability limitation of VILA, we augment additional experiments based on another SOTA Video-LLM, i.e., LLaVA-One-Vision-7B. We vary the number of frames (while keeping the candidate frame number M = 128). The results on the Video-MME (without subtitles) benchmark are presented as follows.

| **_LLaVA-One-Vision-7B_** | **2 Frames** | **4 Frames** | **8 Frames** | **16 Frames** | **32 Frames** | **64 Frames** | **128 Frames** |
|---------------------------|--------------|--------------|--------------|---------------|---------------|---------------|----------------|
| **Uniform Sampling**      | 44.3         | 49.6         | 53.3         | 57.2          | 58.2          | 58.0          | 58.2           |
| **Frame-Voyager**         | 51.4         | 54.8         | 57.5         | 59.2          | 60.4          | 60.5          | 58.2           |
| **Gains**                 | +7.1         | +5.2         | +4.2         | +2.0          | +2.2          | +2.5          | Same Input     |

It is observed that Frame-Voyager shows **consistently better performance when varying the number of frames from 2 to 64**. While the uniform sampling method plateaus at 32 frames (58.2), our Frame-Voyager continues to improve up to 64 frames. Notably, Frame-Voyager achieves with 16 frames (59.2) what uniform sampling requires 32 or more frames to approach (58.2).

[1] VILA: On Pre-training for Visual Language Models. CVPR 2024

[2] ​​LLaVA-OneVision: Easy Visual Task Transfer. ArXiv 2024.

[3] Long Context Transfer from Language to Vision. ArXiv 2024.

---

### Author Response · Authors · 2024-11-20
**General Response of First Round Rebuttal (3/4)**

**3.** ```[qbz8, ULTS]``` **Efficiency & Scalability on Data Collection**: How can we make our data collection pipeline more cost-efficient at scale?

First, we would like to highlight that our data collection process is automated and **one-off**, allowing the resulting data (generated by VILA-8B) to be **directly used** by other Video LLMs, i.e., results of VILA-40B in Table 1 (Lines 292-293) and results of LLaVA-One-Vision-7B in the “1. Generalization on Different Backbones” of this general response. We will open-source our algorithm and also the generated dataset after the paper's acceptance.

Next, for rebuttal, we implement two straightforward approaches to address efficiency concerns and reduce computational resource consumption.

* **Method (1)**: We attempt some optimization strategy on our data collection process, revealing the potential for efficiency improvements. Specifically, we conduct dynamic pruning by filtering combinations based on **frame-to-frame similarity** and **temporal proximities among frames**. This strategy helps us discard the frame combinations containing multiple similar and redundant frames. It reduces the average number of combinations per sample to **14% on the VideoChatGPT dataset with processing time reduced to ~5 hours using 32 A100 GPUs** and to **27% on the NextQA dataset with ~15 minutes using 8 A100 GPUs**.

* **Method (2)**: Following the Reviewer ```qbz8```'s suggestion, we conduct experiments by using CLIP to identify relevant frames. Building upon Method (1), we use CLIP to compute the similarity between all frames and the question, and rank them accordingly. We mark lower-ranked frames as irrelevant.  We believe that only a small portion of video frames directly relate to the question, while the majority of frames are irrelevant. As a result, most frame combinations only containing irrelevant frames can be discarded during the training process. Based on this pruning strategy, we filter out most combinations consisting of only irrelevant frames while retaining a few as low-ranking training samples. **This approach can further reduce data collection time by 60-70%**.

The comparison among the original version data collection, +Method (1), and +Method (1) & (2) is shown below. The backbone model is kept as VILA-8B. **The results demonstrate that the data construction time of Frame-Voyager could be largely reduced to just 4.4% of the original version, while maintaining comparable performance across all benchmarks**. We will include these results in the revised version, and we leave the exploration of more sophisticated efficiency improvements to future work.

| **_Training Data Construction_** | **GPU Hours (VideoChatGPT + NextQA)** | **Time Costs (%)** | **Video-MME (%)** | **MLVU (%)** | **NextQA (%)** | **ActivityNetQA (%)** |
|-------------------------|---------------|---------------------|---------------|----------|------------|-------------------|
| **Original Version**    | 1280+8        | 100%                | 50.5          | 49.8     | 51.1       | 51.6              |
| **+Method (1)**          | 160+2         | 12.6%               | 50.6          | 50.0     | 50.8       | 52.0              |
| **+Method (1) & (2)**        | 56+0.8        | **4.4%**                | 50.5          | 49.7     | 51.0       | 51.9              |

---

### Author Response · Authors · 2024-11-20
**General Response of First Round Rebuttal (4/4)**

**4.** ```[pMFu, qbz8]``` **Overall Efficiency**: How much additional computational overhead is required for training and inference?

We conduct a comprehensive analysis of running time, model parameters, and latency comparisons between the baseline (uniform sampling) and our method, as presented in Appendix C (Lines 219-226).
* Parameter size: our method adds only several MLPs, increasing the total parameter count by just a negligible 0.2% compared to the original model. This minimal parameter increase translates to a modest impact on training and inference resources.
* Training efficiency: our Frame-Voyager module requires 64 GPU hours for separate training, representing only 1.25% additional training time compared to the original model's 5.1k GPU hours as in [1].
* Inference speed: we evaluate latency using 100 randomly sampled examples from Video-MME. Our method averages 1.696 seconds per example, and the baseline is 1.329 seconds per sample. It shows a 27.6% increase in inference time.
* Additionally, we add the memory usage comparison based on the Reviewer ```pMFu```’s suggestion. The memory footprint remains similar: our Frame-Voyager uses 23,069 MiB and the baseline uses 22,425 MiB, making a difference of 2.9%.

|                        | **Parameters Overhead** | **Training Hours** | **Latency** | **Memory** |
|------------------------|-------------------------|--------------------|-------------|------------|
| **VILA**       | -                       | 5.1k GPU Hours     | 1.329s      | 22,425 MiB |
| **VILA+Frame-Voyager** | +0.2%                   | 64 GPU hours       | 1.696s      | 23,069 MiB |

These metrics demonstrate that Frame-Voyager only introduces a tiny computational overhead. The minimal increases in parameter count and memory usage suggest that our method scales efficiently while achieving superior video understanding capabilities.

[1] VILA: On Pre-Training for Visual Language Models. CVPR 2024.

---

### Author Response · Authors · 2024-12-01
**General Response and Revision of Manuscript**

We sincerely thank the reviewers for their valuable feedback!

In this post:

*  We summarize and highlight the key strengths of our work as noted by the reviewers.
*  We summarize the changes to the updated PDF document (marked in blue). Additional experimental results after the PDF revision deadline will be added in the final version.


****(1) Key Strengths****
* **Motivation**
  - ```pMFu```: "*reasonable extension from previous keyframe selection work*"
  - ```6u8S```: "*motivation is clear and easy-to-follow*"
  - ```qbz8```: "*a critical challenge in video understanding…making this problem a key focus for advancing the field*"

* **Method**
  - ```pMFu```: "*good contribution to the keyframe selection in video-language studies*"
  - ```qbz8```: "*novel formulation of frame selection as a ranking problem is compelling…offering a fresh and efficient perspective on the task*"
  - ```ULTS```: "*an efficient keyframe data construction method*", "*a novel approach for improving Video Large Language Models*", "*a valuable tool for video understanding*"

* **Experiments & Results**
  - ```pMFu```: "*extensive experiments across diverse popular benchmarks to show their effectiveness*"
  - ```6u8S```: "*analysis is comprehensive*"
  - ```ULTS```: "*FRAME-VOYAGER achieves significant performance in several VQA datasets*, *strong performance across multiple datasets proving the robustness of the approach*"

* **Presentation**
  - ```6u8S```: "*presentation of the paper is of high quality*"
  - ```qbz8```: "*well-written, with a clear explanation of the motivation and methodology*"



****(2) Changes to the PDF****

* **Section 2 Related Work**
  - ```pMFu```: Adding related works in frame selection and video grounding.
  - ```6u8S```: Adding related works leveraging LLM agents.

* **Section 3 Frame-Voyager**
  - ```pMFu```: Adding the explanation why our loss-based desigen is able to avoid language prior shortcuts in Section 3.1.

* **Section 4 Experiments**
  - ```pMFu,6u8S```: Augmenting results of Frame-Voyager on LLaVA-One-Vision in Table 1.
  - ```ULTS```: Modifying the results on short video benchmarks with higher number of frames in Table 1.
  - ```pMFu,6u8S,qbz8```: Adding more frame selection baselines in Table 2.
  - ```pMFu,6u8S,qbz8,ULTS```: Frame-Voyager generalized to more frames with LLaVA-One-Vision in Figure 3b and related explanation in Research Question 3.
  - ```qbz8,ULTS```: A new Research Question 7 elaborating the efficiency and scalability of Frame-Voyager when processing more frames.

* **Appendix**
  - ```pMFu,6u8S,qbz8```: Updating implementations for the new added frame selection baselines in Appendix A.
  - ```pMFu,qbz8```: Adding extra memory usage of Frame-Voyager in Appendix C.
  - ```ULTS```: Adding results on three additional benchmarks in Appendix D.
  - ```6u8S,qbz8```: Dynamic frame selection experiment in Appendix F.
  - ```qbz8,ULTS```: Optimization of data collection pipeline in Appendix G.
  - ```pMFu```: Case study on the ranking data for training in Appendix H.

* **References**
  - ```[1]``` "On Calibration of Modern Neural Networks". ICML 2017. Guo et al.
  - ```[2]``` "Self-Adaptive Sampling for Accurate Video Question Answering on Image Text Models". Findings of NAACL 2024. Han et al.
  - ```[3]``` "Language Models (Mostly) Know What They Know". arXiv 2022. Kadavath et al.
  - ```[4]``` "MVBench: A Comprehensive Multi-modal Video Understanding Benchmark". CVPR 2024. Li et al.
  - ```[5]``` "TimeCraft: Navigate Weakly-Supervised Temporal Grounded Video Question Answering via Bi-directional Reasoning". ECCV 2024. Liu et al.
  - ```[6]``` "EgoSchema: A Diagnostic Benchmark for Very Long-form Video Language Understanding". NeurIPS 2023. Mangalam et al.
  - ```[7]``` "VideoAgent: Long-form Video Understanding with Large Language Model as Agent". ECCV 2024. Wang et al.
  - ```[8]``` "ViLA: Efficient Video-Language Alignment for Video Question Answering". ECCV 2024. Wang et al.
  - ```[9]``` "VideoTree: Adaptive Tree-based Video Representation for LLM Reasoning on Long Videos". arXiv 2024. Wang et al.
  - ```[10]``` "STAR: A Benchmark for Situated Reasoning in Real-World Videos". NeurIPS 2021. Wu et al.
  - ```[11]``` "Can I Trust Your Answer? Visually Grounded Video Question Answering". CVPR 2024. Xiao et al.

---

### Meta-Review · Area_Chair_8q2Z · 2024-12-21

**Metareview:**

Frame selection is a popular paradigm to improve the computational cost of video-language models. This paper improves upon prior works like Sevila by not choosing frames independently, but by considering temporal relations among frames (the authors address the combinatorial search space of this task by phrasing it as ranking instead).

The rebuttal and revised version of the paper addressed most of the reviewers' initial concerns, and each reviewer is now positive about the paper. Therefore, the decision is to accept the paper.

**Additional Comments On Reviewer Discussion:**

Please see above. The rebuttal and revised version of the paper addressed most of the reviewers' initial concerns, and each reviewer is now positive about the paper.

---

### Decision · Program_Chairs · 2025-01-22

Accept (Poster)